# *Lactobacillus iners* dominates the vaginal microbiota of healthy Italian women of reproductive age

E. Vinerbi,[1] F. Chillotti,[1] A. Maschio,[1] S. Lenarduzzi,[2] S. Camarda,[3] F. Crobu,[1] D. V. Zhernakova,[1] V. Lo Faro,[1] G. Beltrame Vriz,[3] S. Incollu,[1] J. Spreckels,[4] N. Kuzub,[4] A. Kadric,[5] R. Gacesa,[6] A. Zhernakova,[4] F. De Seta,[2,3] D. Mazzà,[2] F. Busonero,[1] M. L. Ferrando,[1] G. Girotto,[2,3] S. Sanna[1,4]

**ABSTRACT**  Large sex hormonal fluctuations are thought to influence vaginal microbiota, but little is known about the impact of small, physiological variations. Here, we tracked changes in vaginal microbiota during four key menstrual cycle phases in 61 healthy, naturally menstruating Italian women from the Women4Health cohort. The microbiota, characterized using a high-depth 16S rRNA amplicon sequencing approach covering four hypervariable regions, was primarily composed of *Lactobacillus* species, with *Lactobacillus iners* being the most abundant (average relative abundance: 40%) and the most prevalent (prevalence: 98%). Individual microbiota were generally stable, but beta diversity was higher during the follicular phase ($P = 0.007$). Only 11 women exhibited compositional shifts, mostly occurring between the follicular and ovulatory phases. Finally, using linear mixed models, we assessed the association between taxa relative abundance and five sex hormones along the menstrual cycle. Among these, 17-beta estradiol showed the largest number of significant associations, linking its increase to a decrease in the relative abundance of taxa that are more common after menopause. Our study highlights specific features of the Italian population and points to the resilience of the vaginal microbiota to physiological hormonal changes. Noteworthy, the observed high abundance of *L. iners* contrasts with previous studies in European populations, challenging its proposed pathogenic role and suggesting distinct microbiota profiles within Europe.

**IMPORTANCE**  The vaginal microbiota plays an important role in women's health, yet we know little about how it responds to normal hormonal fluctuations. In this study, we followed 61 healthy Italian women over a natural menstrual cycle to explore microbiota changes across different hormonal phases. We found that *Lactobacillus iners* was the most common species, unlike previous findings in Northern Europe, suggesting population-specific patterns. The common hypothesis that *L. iners* is invariably linked with poor health is called into question by our findings. They emphasize the importance of considering population context and hormonal status when assessing vaginal health. The vaginal microbiota was generally stable, with only a few changes observed between the follicular and ovulatory phases. When evaluating the association between five sex hormones and taxa abundances, we found that 17-beta estradiol levels had the largest number of significant associations. These highlight an association between increased levels of 17-beta estradiol and increased relative abundance of rare bacterial taxa rather than dominant species like *Lactobacillus*. Our findings help define what constitutes a "healthy microbiota" in generally healthy Italian women of reproductive age and may inform future strategies for diagnosing or preventing women's health conditions.

**KEYWORDS**  vaginal microbiota, *Lactobacillus iners*, *Lactobacillus crispatus*, Italian women, women's health, sex hormones, menstrual cycle

Address correspondence to S. Sanna, serena.sanna@cnr.it, or G. Girotto, giorgia.girotto@burlo.trieste.it.

E. Vinerbi, F. Chillotti, A. Maschio, and S. Lenarduzzi contributed equally to this article. The author order was determined in order of increasing seniority.

The authors declare no conflict of interest.

See the funding table on p. 19.

The vaginal microbiota refers to the community of microorganisms that colonize the vagina. Unlike microbiota from other mucosal sites (i.e., the gastrointestinal tract), it has lower diversity, and it is dominated by a few species, primarily *Lactobacillus* spp., which play a key role in regulating genital health. Producing lactic acid and antimicrobial molecules, these bacteria reduce the risk of colonization by other microorganisms, including potential pathogens involved in infections and diseases of the vaginal tract (1–8). In general, the prevalence of the *Lactobacillus* genus in healthy women increases with puberty and remains stable until menopause, except during vaginal infections (9).

The most common species of *Lactobacillus* are *Lactobacillus iners*, *Lactobacillus crispatus*, *Lactobacillus gasseri*, and *Lactobacillus jensenii*. Earlier studies have shown that a vaginal microbiota dominated by *L. crispatus* is associated with a healthy status, while *L. iners* is often isolated during infections, and therefore a microbiota dominated by it is frequently considered unhealthy. However, this hypothesis originated from studies with small sample sizes, and a direct pathogenic role for *L. iners* has not been proved (2, 10, 11). Likewise, a vaginal microbiota dominated by other anaerobic bacteria, such as *Gardnerella vaginalis* (classified as *Bifidobacterium vaginale* and other *Bifidobacterium* spp. in Genome Taxonomy Database [GTDB] nomenclature [12])*, Prevotella*, and *Fannyhessea*, has been associated with clinical conditions of the female genital tract, such as bacterial vaginosis and vulvovaginal atrophy, due to the virulence potential of some strains (13, 14).

The pathogenetic role of *L. iners* remains under debate since its prevalence can vary substantially among different populations (3, 15–18). The vaginal microbiota of African and Hispanic women during the reproductive age is dominated by *L. iners*, while it is much rarer in Northern European women, with *L. crispatus* being the most abundant and prevalent *Lactobacillus* species (2, 15). The reasons for this north-to-south gradient in the prevalence of *L. crispatus* and *L. iners* are still unknown. It is worth noting that these two species can co-exist in the same microbiota; therefore, it is unlikely they are antagonists (15).

The vaginal microbiota has been considered an overall stable ecosystem, although two recent studies conducted on short-term (42 days) and medium-term (18 months) longitudinal follow-up have shown that variations may occur in a small fraction of women (2, 19). These studies evaluated changes in the so-called Community State Types (CSTs), a method that classifies the vaginal microbiota into five groups depending on the composition and the most abundant species (3, 20). In the shorter 42-day study, only microbiotas classified as CST-I (dominated by *L. crispatus*) and CST-IV (dominated by anaerobic bacteria) remained stable, while variations were observed in other CSTs. In the longer 18-month longitudinal study, vaginal microbiota of CST-I and CST-III (dominated by *L. iners*) remained stable, while CST-IV microbiota displayed lower temporal stability (approximately 6 months). The underlying causes of these variations, how they occur in populations with different *Lactobacillus* spp. prevalences, and which species may be responsible for these changes in CSTs are not yet known.

It is worth noting that none of these studies have characterized the association between compositional changes in the microbiota and specific fluctuations in sex hormones.

Here, we present vaginal microbiome composition and its dynamics in 61 healthy women from the Italian population cohort Women4Health (W4H), an observational longitudinal study centered on one menstrual cycle, set up to explore the interactions between sex hormones, vaginal and intestinal microbiome, and lipid and glucose metabolism in women (21). We characterized vaginal microbiome composition and evaluated dynamics at four time points pivotal for the menstrual cycle: follicular (F), ovulatory (O), early luteal (EL), and late luteal (LL) phases (Fig. 1). We also evaluated host factors associated with diversity and identified sex hormones that are mostly associated with the observed dynamic changes.

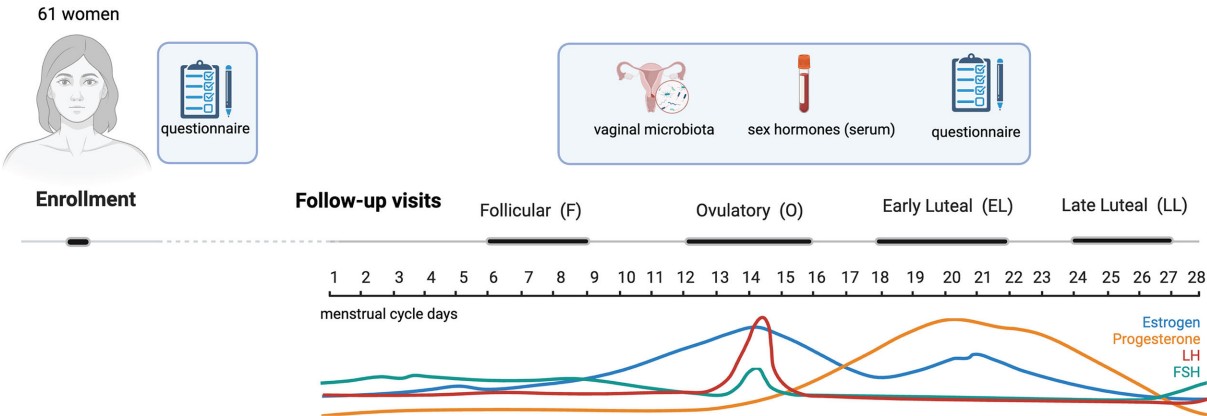

**FIG 1** Schematic representation of data and biological samples collection. Schematic representation of data collected and analyzed in this work. For a description of additional data and samples being collected in the Women4Health study, see reference 21. Created in BioRender. https://BioRender.com/rcl99h9.

## MATERIALS AND METHODS

### The Women4Health cohort

In this study, we analyzed the vaginal swabs, metadata, and sex hormone levels for the first 61 women enrolled in the Women4Health cohort, an ongoing short-term longitudinal study aiming to recruit up to 300 healthy women and to follow them up through a natural menstrual cycle. The W4H study design has been previously described (21). Briefly, women of European origin aged 18–45 years, who were not using hormonal contraceptives, did not have a diagnosis of gynecological, metabolic, or gastrointestinal diseases, and reported the absence of urogenital symptoms, were enrolled in the study at the Institute for Maternal and Child Health–IRCCS Burlo Garofolo in Trieste, Italy. Upon inclusion and signature of an informed consent form, electronic questionnaires were used to gather data on family history, clinical and anthropometric measurements, lifestyle, and diet. They were then scheduled for four weekly follow-up appointments to hand in self-collected biological samples, including a vaginal swab, and to undergo a blood withdrawal. Moreover, they were given an additional questionnaire to collect information related to their health status and lifestyle in the days preceding the appointment. Weekly follow-up visits were scheduled according to pivotal phases of the menstrual cycle: the first at the follicular phase (between days 6 and 9), the second at the ovulatory phase (between days 12 and 16), the third at the early luteal phase (between days 18 and 22), and the fourth at the late luteal phase (between days 24 and 27) (Fig. 1). Blood samples were processed within 1 hour after collection. In serum, we measured five sex hormones (17-beta estradiol, luteinizing hormone, follicle-stimulating hormone, progesterone, and prolactin). During the first visit, we also measured testosterone and thyroid parameters (thyroid-stimulating hormone and thyroxine [$T_4$]) to exclude endocrine disorders. Hormones were measured using 1.5 mL serum on the ADVIA Centaur CP Immunoassay System instrument from Siemens Healthineers. Descriptive statistics for the 61 women participating in this work are presented in Table S1.

### Vaginal swab collection, processing, and microbial DNA extraction

Vaginal swabs were self-collected by volunteers before each follow-up visit (the same day or the day before) using the OMNIgene-VAGINAL kits (OMR-130, DNA Genotek), stored at room temperature, and handed at IRCCS Burlo Garofolo within 24 hours, where kits were promptly stored, as recommended by the manufacturer, and shipped to the Institute for Genetic and Biomedical Research-National Research Council within 21 days. DNA was extracted from swabs with QIAamp PowerFecal Pro DNA kits (QIAgen) using a custom semiautomated protocol on a QIAcube platform. Prior to DNA extraction, a

bead-beating step on a TissueLyser II (QIAgen) at 7.5 Hz for 5 min was performed to improve lysis. The composition of vaginal microbiota was established by sequencing amplicons of the V3–V4 and V7–V9 regions of the 16S rRNA gene, along with the ITS1 gene of fungi. Library construction was performed with a two-step PCR protocol and dual-barcoding strategy (QIAseq 16S/ITS Region Panel kits, QIAgen). Sequencing was carried out with the MiSeq sequencing system (Illumina) targeting an average of 100,000 250-bp paired-end reads per sample. Samples were processed in three different batches, each including one QIAseq 16S/ITS Smart Control (QIAgen), a synthetic DNA construct used both as a positive control for library preparation and as a control for the identification of contaminants, and a PCR negative control (distilled sterile water). Data obtained were demultiplexed using Qiagen's GeneGlobe software.

## Quality control process

All demultiplexed FastQ files underwent quality control screening before bioinformatic analysis. FastQC version 0.12.1 (see https://www.bioinformatics.babraham.ac.uk/projects/fastqc/) and MultiQC version 1.22.2 (22) (see https://github.com/MultiQC/MultiQC) tools were used to detect outliers based on read quality. The trimmomatic tool version 0.39 (23) (see https://github.com/timflutre/trimmomatic?tab=readme-ov-file) was used for trimming low-quality paired-end reads. We used the options SLIDINGWINDOW:4:25 to evaluate whether the average quality score was at least 25 per four bases; if not, this part of the read was trimmed. The option MINLEN:50 was used to remove reads shorter than 50 bp. After the trimming process, samples were re-analyzed with FastQC and MultiQC to evaluate their post-process quality.

We first processed 185 samples, 4 smart controls, and 3 negative controls in two plates and decided to remove (i) all samples that had less than 80,000 reads as input before the trimming process and/or less than 60,000 reads survived post-trimming ($N$ = 10 samples), (ii) one sample with a very high number of internal transcribed spacer (ITS) read mapping on *Sporobolomyces* spp., a known environmental contaminant, and (iii) samples whose libraries were prepared together with a negative control that showed contamination (>500 reads) ($N$ = 10) (Table S2). These 21 samples were repeated on a third plate, containing an additional 28 samples, 1 smart control, and 1 negative control. The same quality control filters were applied, and only 1 sample was removed from plate 3. Finally, one sample that showed 1.5 million reads and 1.3 million post-trimming reads as input, due to an error in library equimolar pooling, was subjected to the subsampling process by randomly selecting 187,000 reads (the average value of reads survived post-trimming for plate 3) through the R package ShortRead version 1.62.0 (see https://kasperdanielhansen.github.io/genbioconductor/html/ShortRead.html).

After all these quality control (QC) steps, a total of 212 samples were left for downstream analyses, with the median number of QCed reads per sample being 138,531 (127,689, 141,450, and 190,975 for plates 1, 2, and 3, respectively), across all sequenced regions. The average number of QCed reads was 148,660.0 for V3V4V7V9 regions and 195.2 for the ITS region.

## Characterization of the vaginal microbiota

To characterize the bacterial composition of the samples, the DADA2 version 3.19.0 package (24) (see https://www.bioconductor.org/packages/release/bioc/html/dada2.html) was used in the R (version 4.4.1) environment. Using the "filterAndTrim()" function, only sequences with a minimum length of 160 bp were selected. With the "removeBimeraDenovo()" function, all chimeras were removed.

The microorganisms were classified up to the taxonomic level of species with the "assignTaxonomy()" and "addSpecies()" functions. The Genome Taxonomy Database version r95 (25) was used to profile a phylogenetically consistent microbial taxonomy (see https://gtdb.ecogenomic.org/stats/r95). Hypervariable regions V3V4 and V7V9 were processed together. Rare amplicon sequence variants (ASVs) resulting from DADA2 were removed; only ASVs that had at least five reads in at least two samples were maintained

for subsequent analyses. To increase taxonomic resolution for the genus *Lactobacillus,* we followed a previously described protocol (15). Using their code and data files provided (see https://github.com/LeberLab/Citizen-science-map-of-the-vaginal-microbiome), the genus *Lactobacillus* was reclassified into different subgenera, allowing the identification of more *Lactobacillus* species. Finally, *Lactobacillus* species names were replaced by the respective subgenus identified (Table S3). Next, we selected ambiguous ASVs—i.e., those ASVs that were assigned by the algorithm as *Lactobacillus* species but belonged to other genera—and uploaded the corresponding sequences on BLAST to identify the most similar species. Since ambiguity remained, we decided to assign these ASVs the label "unclassified" (Table S4).

To characterize the fungi composition of the samples, the ITS fastQ files were processed with the DADA2 version 3.19.0 (24) (see https://www.bioconductor.org/packages/release/bioc/html/dada2.html) package on R (version 4.4.1). To this end, the "filterAndTrim()" function was set as "minLen = 50" and "maxLen = 200." We profiled samples with the Unite database version 10.0 (26) (see https://doi.org/10.15156/BIO/2959330).

We derived global metrics of bacterial microbiome composition, such as alpha and beta diversity, using the vegan package version 2.6-4 (27) (see https://github.com/vegandevs/vegan/). Alpha diversity was calculated based on taxa counts according to the Shannon-Weiner (H) and Simpson (D) indices. Beta diversity was calculated according to the Bray-Curtis dissimilarity index using taxa relative abundances and according to Euclidean distance using taxa relative abundances normalized using the centered log ratio (CLR) transformation. The Euclidean distance was used in addition to Bray-Curtis because it allowed us to use normalized and corrected data (Table S5). To perform CLR transformation, we used our own function that calculates the transformation in parallel among taxa, using the Parallel package version 4.4.1 on R (see https://www.rdocumentation.org/packages/parallel/versions/3.6.2).

In addition, we classified our samples according to the CSTs definition of the VALENCIA algorithm (20) (see https://github.com/ravel-lab/VALENCIA). Toward this end, it was necessary to manually modify the taxonomy names according to those required by VALENCIA (Table S6). To identify which ASV could be attributed to *Gardnerella vaginalis*, given the different nomenclature in GTDB, we ran DADA2 again with the SILVA database (version 138.1), where the nomenclature of species aligns with that used in the VALENCIA algorithm (Fig. S1). Specifically, we identified 46 ASVs assigned to the genus *Bifidobacterium* using GTDB version r.95. Among these, 15 ASVs classified as *B. vaginalis* (six ASVs) and *B.* unclassified (nine ASVs) corresponded to *Gardnerella vaginalis* when annotated using the SILVA database. These 15 ASVs were renamed accordingly and used in the VALENCIA algorithm, while the remaining ASVs were discarded.

Finally, we used the t-SNE algorithm (28) to visualize the vaginal microbiome composition in a two-dimensional space and used color-coding labels to describe the observed variation in terms of taxa abundance as well as CSTs (Fig. S2). For this analysis, we employed the t-SNE algorithm to CLR-transformed data using the R package *Rtsne* version 0.17 (29) (see https://cran.r-project.org/web/packages/Rtsne/index.html).

Finally, we profiled microbiotas using the same pipeline but subsampling 30,000 (and 60,000) survived reads per sample using the R package ShortRead (see https://kasperdanielhansen.github.io/genbioconductor/html/ShortRead.html).

## Statistical analyses

### Identification of factors associated with beta diversity

We evaluated the impact on beta diversity of technical confounders, characteristics of volunteers (as recorded in questionnaires), and of sample collection. Using the vegan package (27) (see https://github.com/vegandevs/vegan/), we first investigated whether the first five Bray-Curtis beta diversity principal coordinate analysis (PCoA) axes were correlated with technical variables, such as microbial DNA concentration, library concentration, and total read counts (survived reads). Each PCoA axis was linearly

adjusted using only the covariates with which it showed a significant correlation. As the data set included longitudinal measurements, independence tests were conducted separately for each visit to ensure independence between observations. Namely, we used the Kruskal-Wallis independence test in R through the function "kruskal.test()" of the stats package version 4.4.1 (30) (see https://doi.org/10.32614/CRAN.package.STAT).

Variables from the questionnaires encompassed both categorical and continuous data related to demographics (e.g., age), anthropometric measures (e.g., body mass index and waist-to-hip ratio [WHR]), lifestyle factors (e.g., smoking and hormonal contraceptive use), clinical history related to women's health (e.g., number of pregnancies), and conditions surrounding vaginal swab collection (e.g., medication or supplement use prior to collection, sexual intercourse within 2 days of collection, swab timing, and stool consistency assessed using the Bristol stool scale). Continuous variables were categorized to allow the application of the independence test. All tested variables are summarized in Table S7.

We also ran PERMANOVA analyses for the significant associations resulting from the independence test, using the adonis2() function from the vegan R package (27) (see https://github.com/vegandevs/vegan/).

### Evaluation of vaginal microbiota dynamics

To evaluate changes in alpha and beta diversity across the four phases of the menstrual cycle, we used the non-parametric Friedman test. To evaluate changes between two phases (pairwise comparisons), we used the non-parametric Wilcoxon pairwise test for the untransformed data, and the parametric pairwise *t*-test for the CLR-transformed data. All these comparative tests were run using the stat package version 4.4.1 (30) (see https://doi.org/10.32614/CRAN.package.STAT). All pairwise comparisons were corrected using the Bonferroni approach.

To evaluate beta diversity changes (both Bray-Curtis and Euclidean distance, adjusted for technical covariates), we used two approaches. First, we calculated mean and median beta diversity across all women within each menstrual cycle phase and compared these across the different phases using the paired Wilcoxon test (Table S8).

In a second step, we calculated the mean and median beta diversity for each woman among her different collections (up to four) and compared the distribution obtained when comparing her first collection to three collections from different women from the other three phases, using again the unpaired Wilcoxon test. All comparisons were corrected using Bonferroni.

We instead used linear models to investigate the dynamics of specific taxa and to assess their association with sex hormones. Specifically, we fitted three groups of linear models to (i) investigate whether bacterial abundances vary across the menstrual cycle, (ii) explore the relationship between hormones and bacterial abundance variations between two consecutive phases, and (iii) explore the overall relationship between hormones and bacterial relative abundances.

For all linear models, taxa were filtered using a prevalence >20% and a mean relative abundance threshold of 0.10% (Table S9).

In the first group of models, we fitted the following Gaussian mixed models:

$$\text{Taxa\_adj} \sim \text{visit\_number} + (1|\text{woman}), \qquad (1)$$

where visit_number is a numeric variable coded as 1, 2, 3, and 4 according to the four phases of the menstrual cycle, and Taxa_adj are the residuals of the linear models:

$$\text{Taxa\_CLR-transformed} \sim \text{covariates}$$

fitted using the function "LinearRegression()" of sklearn package version 1.5.2 in Python version 3.10.12 (31) (see https://jmlr.csail.mit.edu/papers/v12/pedregosa11a.html). The placeholder "covariates" here means that we systematically

considered several sets of covariates based on the results of independence tests and technical variables to evaluate if these changes could be confounded by a specific covariate. We opted for stepwise models with an increasing number of covariates, rather than running a model with the full set of covariates, given the limited sample size of the study. A total of 10 models were considered (Table S10).

From equation 1, we used the regression coefficients and the *P*-values for the variable Visit_number to identify taxa with significant changes. We used permutations to derive an empirical *P*-value and evaluate robustness of significant results ($P < 0.05$). We permuted the column containing taxa abundances and re-ran the model 1,000 times. The proportion of "fake" *P*-values less than or equal to the original *P*-value was calculated to provide the empirical *P*-value. In addition, false discovery rate (FDR) was instead used to adjust *P*-values for multiple testing (original *P*-values, not empirical *P*-values), with the function "p.adjust()" of the stats package and method "BH" (32).

In addition, we calculated the interclass correlation coefficient (ICC) for the model where bacteria were adjusted for the technical covariates (model i above). This parameter quantifies the proportion of total variance attributable to between-subject variability, reflecting the stability of bacterial abundance across different time points within subjects. The ICC was calculated using the function "icc()" from the package performance version 0.12.4 (32) (see https://doi.org/10.32614/CRAN.package.performance).

In the second group of models, we again used linear mixed models to investigate if changes in bacterial relative abundance between two phases were associated with changes in sex hormones.

First, we removed outlier values in the sex hormone variables exceeding four standard deviations from the mean. Second, we calculated the differences between two consecutive phases of both sex hormones and bacterial relative abundances, and then we normalized these differences using the function "ordernorm()" of the R package bestNormalize version 1.9.1 (33) (see https://cran.r-project.org/web/packages/bestNormalize/index.html).

The following Gaussian mixed models were fitted:

$$(\mathrm{Hormone}_{t+1} - \mathrm{hormone}_t) \sim (\mathrm{taxa\_adj}_{t+1} - \mathrm{taxa\_adj}_t), \tag{2}$$

where *t* represents the menstrual cycle phase, and taxa_adj is the CLR-transformed abundance adjusted by technical covariates, calculated as the residuals of the models:

$$\mathrm{Taxa\_CLR\_transformed} \sim \mathrm{Qubit\_DNA} + \mathrm{Qubit\_Library} + \mathrm{Total\_counts}, \tag{3}$$

where Qubit_DNA is the DNA concentration, Qubit_Library is the library concentration, and Total_counts is the total number of survived reads.

In the third group of models, we investigate relationships between taxa relative abundances and sex hormone changes across the entire menstrual cycle by fitting the following:

$$\mathrm{Hormone} \sim \mathrm{visit\_number} + \mathrm{taxa\_adj} + (1|\mathrm{woman}) + \mathrm{pregnancy\_category}. \tag{4}$$

In both equations 2 and 4, we used the same strategies described above for equation 1 to derive empirical *P*-values and to correct *P*-values for multiple testing. To fit all linear mixed models in equations 1, 2, and 4, we used the lme4 version 1.1.35.5 (34) (see https://doi.org/10.32614/CRAN.package.lme4) and the lmerTest version 3.1.3 (32) R packages (see https://doi.org/10.32614/CRAN.package.lmerTest).

### Shotgun metagenomics sequencing

A total of 10 samples with different microbiome compositions were selected for whole-genome sequencing (WGS). We used the beta diversity matrix to maximize dissimilarity among selected samples, as follows. The first three selected samples

were those showing a high relative abundance of *L. iners, L. crispatus,* and *L. gasseri,* respectively, and had the maximum beta diversity among them. Then, we selected seven samples showing the maximum distance from the first three samples. Library preparation and sequencing was performed at Prebiomics srl on the Illumina NovaSeq 6000 sequencing system, and microbiome profiling was carried out using MetaPhlan 4 (35). An average of 3 million bacterial reads per sample was obtained (Table S11a). Relative abundances from the two data sets (16S and WGS) were compared for taxa with abundances > 0 in at least one of the samples. Spearman's correlation test was used to check the correlations between relative abundances of taxa detected by both approaches and for which at least three samples had non-zero value (Table S11b).

## RESULTS

### Vaginal microbiome composition

A total of 61 women contributed 213 vaginal swabs to this study, of which 212 passed QC as described in Materials and Methods. Bacterial composition was profiled using previously described protocols (see Materials and Methods). At the genus level, *Lactobacillus* was the most common genus, followed by *Bifidobacterium* and *Streptococcus*. The two most abundant *Lactobacillus* spp. were *L. iners* and *L. crispatus* (Fig. S2) and were identified in almost all samples (prevalence > 90%). When restricting to samples with a relative abundance > 60%, *L. iners* was detected in 43.4% of the samples (corresponding to 62.2% of women), whereas *L. crispatus* was detected in 34.9% (55.7% of women). In addition to *Lactobacillus* spp., we identified other species, such as *Bifidobacterium* spp. and *Fannyhessea vaginae* (Fig. S3 and Table S9). Of note, some of the ASVs classified as *Bifidobacterium* spp. by the GTDB used in this study would be classified as *Gardnerella vaginalis* according to other nomenclatures (see Materials and Methods and Fig. S1).

Overall, vaginal microbiota remained stable throughout the four menstrual cycle phases investigated. There were no significant differences in alpha diversity across the phases, as measured by the Shannon (H) and Simpson (D) indices (Fig. S4 and Table S12). The average beta diversity was instead higher in the follicular phase compared to the ovulatory phase ($P_{\text{Wilcoxon}}$ = 0.009 and $P_{t\text{-test}}$ = 0.007) and early luteal phase ($P_{\text{Wilcoxon}}$ = 0.04 and $P_{t\text{-test}}$ = 0.03) (Fig. 2A; Table S8). Similar results were obtained when using median instead of average (Fig. S5 and Table S8) or the overall distribution (Fig. S6 and Table S13). Nevertheless, these changes were not significant and did not lead women's microbiota to align to a similar composition during a particular phase. In fact, the average beta diversity of a woman's microbiota across all her samples was lower than the average obtained when using random samples from other women ($P$ value ≪ 0.05 for all comparisons), indicating that microbiota is largely determined by individual factors, rather than by menstrual phase (Fig. 2B). We investigated the impact of host factors and described these in the next paragraph.

Analyses of dynamic changes could not be performed for the mycobiome due to the limited number of samples for which the mycobiome could be profiled. Although we sequenced the ribosomal internal transcribed spacer (ITS) region in all 213 samples, only 25 samples had at least 200 reads, a cutoff deemed necessary to obtain fungi classification (Fig. S7 and Table S14). Twenty-five fungal species were identified, the most prevalent being *Candida albicans*, a fungus associated with vaginal infections (candidiasis) but also frequently found as part of the normal vaginal microbiota. The average number of reads on the eight women with the presence of *Candida albicans* and at least 200 fungal reads was low (749 compared to 92,832 reads on *Lactobacillus* spp. in the same women) (Table S14), so the presence of this fungus is unlikely to be a sign of infection. Of note, we excluded at enrollment all women who reported currently having vaginal infections or urogenital symptoms. The relatively low number of reads overall detected in the ITS region may be due to the use of a protocol and kit optimized for bacterial rather than fungal DNA extraction, as reported in previous studies (36).

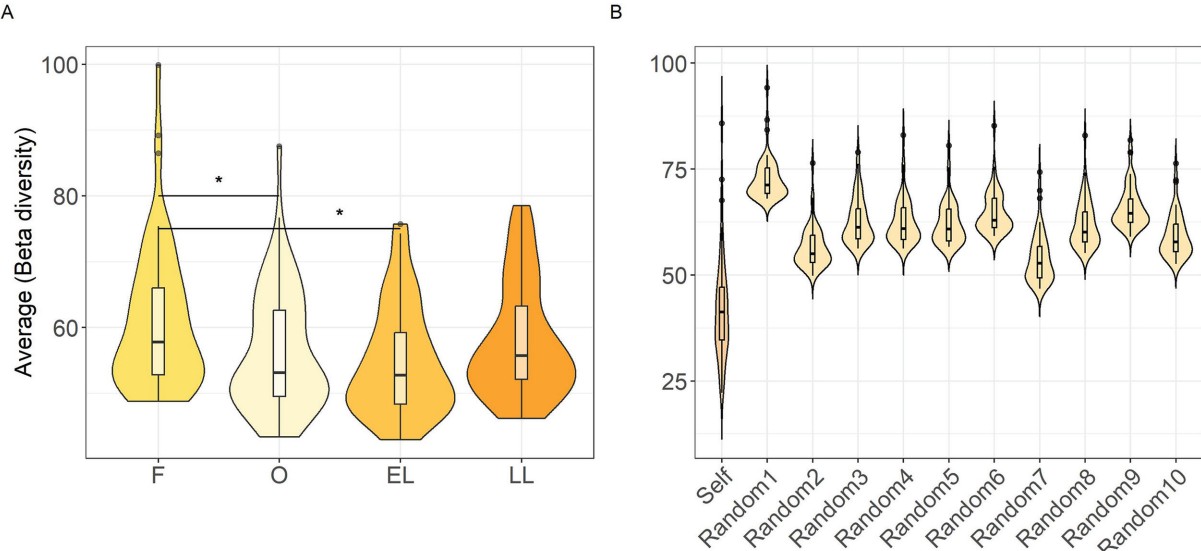

**FIG 2** Average beta diversity across menstrual cycle phases. (A) Average beta diversity across the four menstrual cycle phases: follicular phase, ovulatory phase, early luteal phase, and late luteal phase. The black line inside the boxplot represents the median, with the lowest and highest values within the 1.5 interquartile range represented by the whiskers. The black lines between violins indicate significant differences (*$P < 0.01$). $P$-values were obtained using paired $t$-test. (B) Average beta diversity within the same woman across phases (Self) and average of beta diversity among different women (Random 1 to Random 10). In both panels, beta diversity was calculated using Euclidean distances on transformed taxa adjusted for technical covariates (see Materials and Methods).

## The impact of host phenotypes

We then investigated the impact on vaginal microbiota of host phenotypes, lifestyle, and information on sample collection by correlating genus-based and species-based beta diversity PCoA axes with available metadata. Among the 12 variables tested, the variable "having had at least one pregnancy" had the highest impact at both genus and species levels, followed by "having used oral hormonal contraceptives in the past" (Table S15). Other variables that were significant, albeit with less impact, include age, waist-to-hip ratio, smoking habits, intercourse in the days preceding the swab, time of collection, and use of medication or supplements (Fig. S8). Intriguingly, using a PERMANOVA test and projecting values in the PCoA space, we observed that the microbiota of women who had at least one pregnancy before participating in the study had a lower likelihood to be dominated by *L. crispatus* ($P = 0.005$), in line with observations from the ISALA study (15), suggesting the existence of a potential lifelong pregnancy signature (Fig. S9). Likewise, using PERMANOVA, we observed that the microbiota of older women is less likely to be dominated by *Lactobacillus* spp. ($P = 0.003$), in line with the observed decline in their abundance during menopause (Fig. S10) (9, 37). Similarly, dominance of *Lactobacillus* spp. is less frequent in women who reported collecting vaginal swabs after feces collection compared to those who collected them prior to feces ($P = 0.02$) (Fig. S11), and in those who collected vaginal swabs in the morning compared to those who collected them during the day ($P = 0.003$) (Fig. S12).

## Frequency of *L. iners* and *L. crispatus* along the menstrual cycle

In contrast to findings from cohort studies on Northern European women (Belgium, Denmark, and Sweden) (2, 15), our study found a greater abundance of *L. iners* than *L. crispatus*, a microbiome composition typically observed instead in Hispanic and African populations (3, 16, 18) (Table 1). Notably, all volunteers have reported absence of urogenital symptoms and/or diagnosis for sexually transmitted diseases or infection, so we can rule out the possibility that the observed abundance of *L. iners* in our samples is due to either vaginal pathologies or infections. We also ruled out the possibility that

**TABLE 1** Prevalence and relative abundance of *L. iners* and *L. crispatus* in our study and other European cohorts[a]

| Study | Country | Sequencing method | N reads (K) | N samples | Age range | Time point | Frequency (%) | | Prevalence (%) | | Average relative abundance (%) | |
|---|---|---|---|---|---|---|---|---|---|---|---|---|
| | | | | | | | CST-I | CST-III | *L. iners* | *L. crispatus* | *L. iners* | *L. crispatus* |
| ISALA | Belgium | 16S (V4) | 25 | 3,000 | 18–73 | Any | 43 | 28 | 72 | 90 | 24 | 38 |
| MiMens | Denmark, Sweden | 16S (V3V4) WGS | – | 49 | 20–28 | Follicular phase | – | – | 63 | 76 | 12 | 59 |
| | | | | | | Ovulatory phase | – | – | 66 | 83 | 14 | 67 |
| i-Predict | France | 16S (V3V4) | 34 | 241 | 18–25 | Any | 41 | 38 | – | – | 38 | 45 |
| W4H | Italy | 16S (V3V4, V7V9) | 155 | 61 | 18–45 | Follicular phase | 25 | 42 | 98 | 95 | 40 | 25 |
| | | | | | | Ovulatory phase | 36 | 38 | 93 | 93 | 37 | 35 |
| | | | | | | Early luteal phase | 35 | 37 | 100 | 98 | 34 | 35 |
| | | | | | | Late luteal phase | 30 | 35 | 100 | 100 | 33 | 31 |

[a]The table reports the statistics from previously published studies on European cohorts and those from our study for *L. iners* and *L. crispatus*. The column Time point indicates whether statistics refer to samples collected in a specific menstrual cycle phase or at any time during menstrual cycle (or menopause). "–" indicates that the corresponding information could not be derived from the original article.

our result is driven by a potential bias from the 16S rRNA gene profiling method, since comparison of the relative abundances of these two *Lactobacillus* spp. in 10 samples showed very high concordance with profiles obtained using the shotgun WGS method (Fig. S13 and Table S11b). Likewise, we also excluded the possibility that this result is driven by the higher number of reads compared to the protocols used in previous studies. In fact, taxa relative abundances were almost identical when using a subsampling approach on our 16S rRNA data to restrict analyses to a maximum of 30,000 or 60,000 reads per sample (Fig. S14).

In our cohort, *L. iners* and *L. crispatus* have a similar prevalence in the follicular phase (98% and 95%, respectively), but the average relative abundance was higher for *L. iners* (average relative abundance was 40% and 25% for *L. iners* and *L. crispatus*, respectively) (Table 1). The difference in average relative abundance between these two species was negligible in the other three phases (absolute difference between percentages: 2.03,–1.27, 2.21 in the ovulatory, early, and late luteal phases, respectively). Interestingly, all microbiota dominated by *L. crispatus* in the follicular phase were stable during the menstrual cycle, while a small fraction (20%) of those dominated by *L. iners* at the follicular phase switched to *L. crispatus*-dominant—or to other species—in the ovulatory or in the late luteal phase (Fig. 3).

In addition, the microbiota of two women dominated by low-prevalent species in the follicular phase became *L. crispatus* dominant in the ovulatory phase. ICC analyses indicated that these two species explained the largest variance of total variability between women (Table S16). While these results suggest a trend for women's microbiota to switch to *L. crispatus* during the ovulatory phase, *L. iners* still remains the most abundant species in this cohort across all time points.

## Vaginal microbiota dynamics

Given these clear-cut changes observed in the main *Lactobacillus* spp., we sought to thoroughly investigate microbiota dynamics along the menstrual cycle. First, we assessed the composition and variations along the menstrual cycle using the CST classification (see Materials and Methods). Overall, we found that CST-V (*L. jensenii* dominated) was the rarest among all samples and phases (3.3%), followed by CST-II (*L. gasseri* dominated) (12.7%) and CST-IV (defined by moderate to high abundance of different anaerobic species, such as *Prevotella* spp. and *G. vaginalis,* and low abundance of *Lactobacillus* spp. [14.1%]) (Fig. S15A). In line with relative abundances, the CST-III (*L. iners* dominated) was the most common (38.2%) and with higher prevalence in the follicular phase, while the second most common was CST-I (31.6%) (*L. crispatus* dominated), whose prevalence increased during the ovulatory phase (Fig. S15B). Most women maintained a stable CST

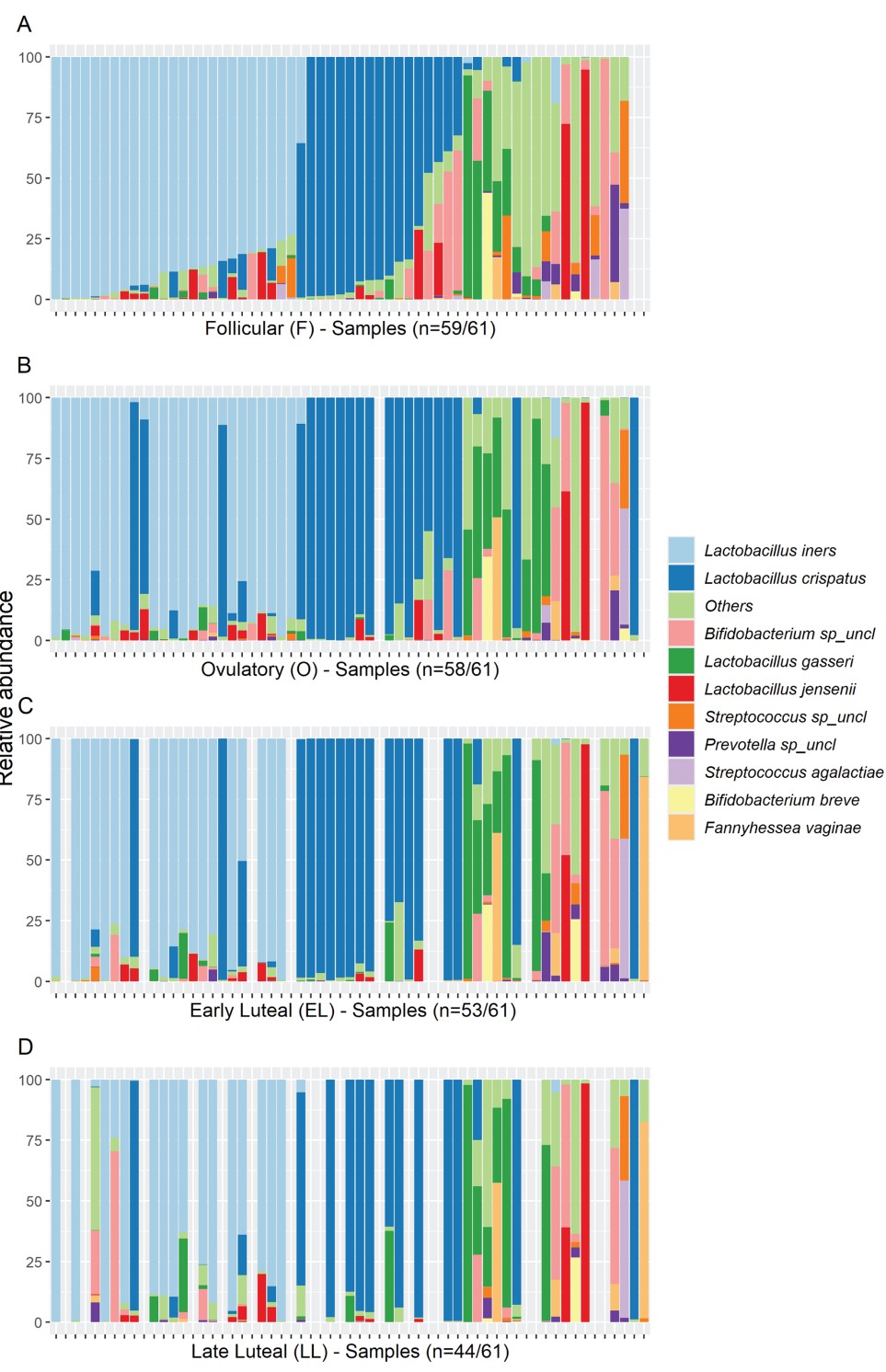

**FIG 3** Distribution of the most abundant taxa across menstrual cycle phases. The figure shows the relative abundance of the 10 most abundant species in the four different phases. (A) Samples are sorted according to the abundance of *L. iners*, *L. crispatus*, and *L. gasseri*, respectively. Samples in panels B–D are ordered as in panel A. The gray bars show missing information for those women in the specific week. In the *x*-axis title, we indicate how many samples had the microbiome profile available. Of note, some of the ASVs classified as *Bifidobacterium* spp. by the GTDB used in this study would be classified as *Gardnerella vaginalis* according to other nomenclatures (see Materials and Methods and Fig. S1).

A

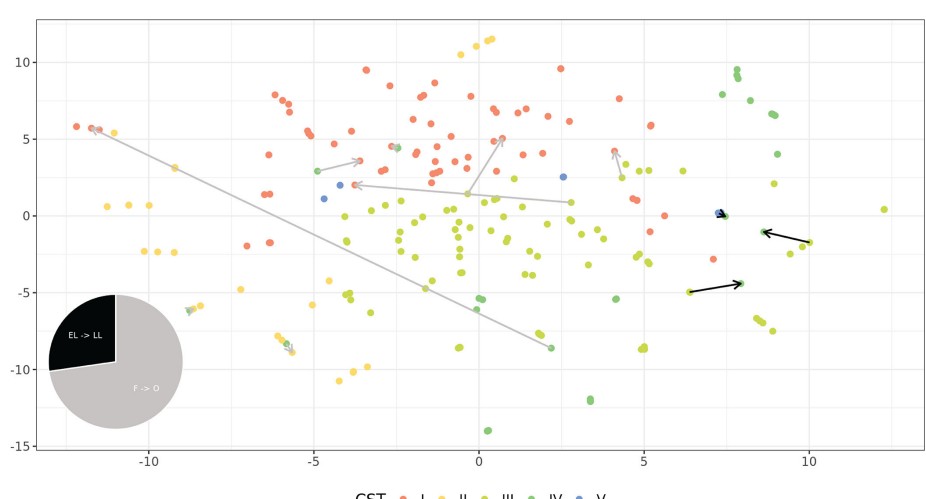

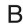

B

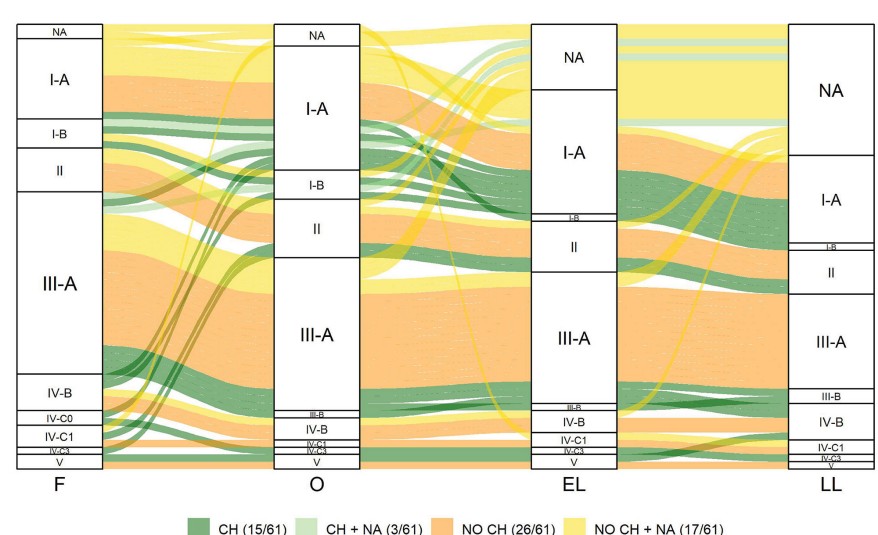

**FIG 4** Community state types across menstrual cycle phases. (A) t-SNE representation of the vaginal microbiome composition, samples are colored according to their CST classification. Arrows highlight instances where a woman changes CST between consecutive phases, with the notation start_phase → final_phase indicated with color coding of the arrow as in the pie chart. In the bottom-left corner, a pie chart shows the proportions of phases where CST changes occur. The legend provides color coding for points according to the assigned CST. (B) Alluvial plot showing subCST changes between phases. The four columns represent the follicular, ovulatory, early luteal, and late luteal phases, respectively. In the legend of the plot, CH: a change occurred in at least one phase, complete data available (15/61); CH + NA: a change occurred in at least one phase, missing samples in some phases (3/61); NO CH: no changes occurred, complete data available (26/61); NO CH + NA: no changes occurred, but missing samples in some phases (17/61).

throughout the menstrual cycle: only 11 out of 61 women changed CST at any time point (18 if we consider changes in subCST) (Fig. 4A). Interestingly, 72% ($N = 8$) of these changes occurred from follicular to ovulatory phases (Fig. 4B). Among them, six were transitions from CST-III ($N = 4$) or CST-IV ($N = 2$) to CST-I. The remaining changes occurred from early to luteal phase, with three women's microbiota switching to CST-IV.

None of the women's microbiota classified as CST-I exhibited CST shifts, indicating the high stability of communities dominated by *L. crispatus*. Of the women who undergo CST

shifts, only one changed CST types more than once (Fig. S16). We were unable to detect association with CST/subCST changes and presence of fungi, given the limited amount of information (only 4 out of 11 women exhibiting shifts in CST had fungi information at the relevant time points). Likewise, we did not detect an association between CST/subCST changes and having had sexual intercourse in the 2 days prior to sample collection ($P >$ 0.05).

Second, we examined the longitudinal changes in taxa relative abundances using linear mixed models and controlling for potential confounding variables, as described in Materials and Methods. We found that the relative abundance of a few bacterial species varies significantly across phases, with the majority remaining stable when different covariates were adjusted. For the genus *Lactobacillus*, *L. iners, L. gasseri,* and *L. jensenii* showed significant variations across most of the corrections applied, while *L. crispatus* was not significant in any of the models (Fig. 5).

Likewise, for the genus *Streptococcus,* two taxa, namely, *S.* unclassified and *Streptococcus agalactiae,* varied significantly across phases regardless of the covariates used. Finally, another bacterial species, *Dialister micraerophilus,* showed significant variation in 3 of the 10 models (Fig. 5).

## Longitudinal changes in bacterial abundance associate with sex hormone levels

We investigated whether these observed significant changes in *Lactobacillus* and *Streptococcus* species between two consecutive phases were associated with changes in sex hormone levels. We found that an increase in the abundance of *L. crispatus* between early luteal and late luteal phases was associated with an increase in progesterone (PROG) levels ($P = 0.03$) (Table S17; Fig. 6A). However, these results were only nominally significant; therefore, larger sample sizes and/or additional cohorts are needed to confirm this finding.

Furthermore, among all taxa with prevalence >20%, we identified several bacteria whose longitudinal fluctuations along the entire menstrual cycle were significantly associated with sex hormone levels, resulting in a total of 14 significant associations (Fig. 6B; Fig. S17 and Table S18). Most of the associations involved 17-beta estradiol (BES17), with some shared between BES17 and LH, including the *Prevotella* genus and species *P. timonensis_A*, and the genera *Finegoldia* and *Anaerococcus*. Only one association was detected with prolactin, namely with the abundance of *Bifidobacterium vaginale*. All these associations were negative, indicating that an increase in sex hormone levels is associated with a decrease in the relative abundance of these bacteria (Fig. 6B; Fig. S17). The number of significant associations decreased to 9 out of 14, when considering permuted *P*-values to evaluate the robustness of results to outliers. Notably, four of these nine associations, all linked to BES17, also meet a stricter significance threshold applied after correcting multiple testing, suggesting that these are likely genuine.

## DISCUSSION

We carried out the first short-term longitudinal study on the vaginal microbiota of non-pregnant, healthy women from an Italian population. In the W4H cohort, we observed a higher abundance of *L. iners* compared to *L. crispatus*, in contrast to earlier studies on cohorts from Northern Europe (2, 15) (Table 1). Of note, a recent study on young French women (19) reported as well a higher prevalence of *L. crispatus*-dominated CST (CST-I) over the *L. iners*-dominated *CST* (CST-III), although the difference in frequency between the two CSTs was less pronounced (40.5% vs 38.1%, while our study has obtained 32% for CST-I and 38% for CST-III) (Fig. S15) compared to observations from cohorts based in Belgium (ISALA study), Denmark, and Sweden (MiMens study) (2, 15). In the French cohort, 70% of women were enrolled in Bordeaux (south of France). Hence, considering these results and the higher prevalence of *L. iners* in African and Hispanic cohorts (3, 16, 17), the results from our study of the Italian population provide evidence for a north-to-south gradient of increased *L. iners* abundance even within Europe.

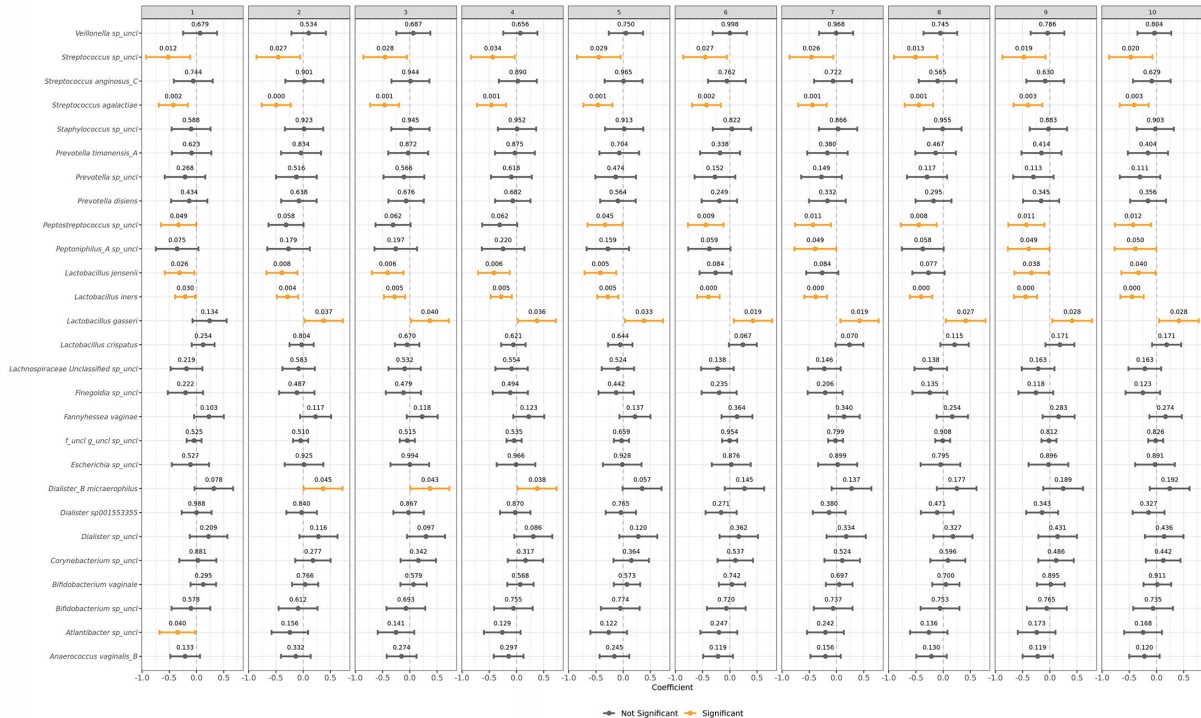

**FIG 5** Longitudinal changes in taxa relative abundance across phases. This figure shows the results of a linear model investigating whether bacterial abundances change across menstrual cycle phases. In each panel, the model coefficients for each taxon are shown with points representing the beta coefficient of the variable time and the bars at the 95% confidence interval of the estimate. The actual *P*-values are shown above each point. Color coding indicates the significance of the associations (*P*-values < 0.05, yellow, black otherwise). Each panel corresponds to the same model using different corrections for the taxa, with the addition of an increasing number of covariates. In particular: model 1 is the basic model with CLR-transformed bacteria, model 2 adds to the previous model the technical covariates, model 3 adds Age, model 4 adds sex_less_than_two_days, model 5 adds Pregnancy_category, model 6 adds Pill_use, model 7 adds Swab_after_feces, model 8 adds Bristol_stool_scale, model 9 adds Swab_morning_or_not, and model 10 adds WHR_ranges. Of note, some of the ASVs classified as *Bifidobacterium* spp. by the GTDB used in this study would be classified as *Gardnerella vaginalis* according to other nomenclatures (see Materials and Methods and Fig. S1).

The reasons for these geographical differences in microbiota composition remain unclear, but results from our study and those from the ISALA study offer insights for speculation. For example, in our study, we found that women who previously had children were less likely to have a microbiota dominated by *L. crispatus*. Likewise, in the ISALA study, having children was negatively correlated with *L. crispatus* and positively correlated with *L. iners*; this variable was the strongest host factor associated—with opposite direction—with the abundance of these two species (15). Geographic differences may thus be related to the delivery mode or post-pregnancy health care practices. Another contributing factor might be related to differences in cultural and hygiene practices, such as the soap type used, the frequency of washing, the use of intimate wipes, and even the type of pads, tampons, or other menstrual supplies (5, 38). While we do not have this information in our cohort, results from the ISALA study indicate that the use of menstrual pads is negatively correlated with *L. crispatus* abundance, while the use of menstrual cups was positively correlated (15). Intriguingly, the ISALA study also reported a positive correlation with *L. crispatus* abundance with the use of hormonal contraceptive methods (combination pill, vaginal ring, or patch). Previously, Tuddenham et al. (39) showed that the proportion of CST I (*L. crispatus*-dominated) was higher among users of oral hormonal contraceptives than the proportion of CST I among non-users, albeit this result was significant only in White and not in African American. No association was instead seen in these two studies between hormonal contraceptive use and *L. iners*.

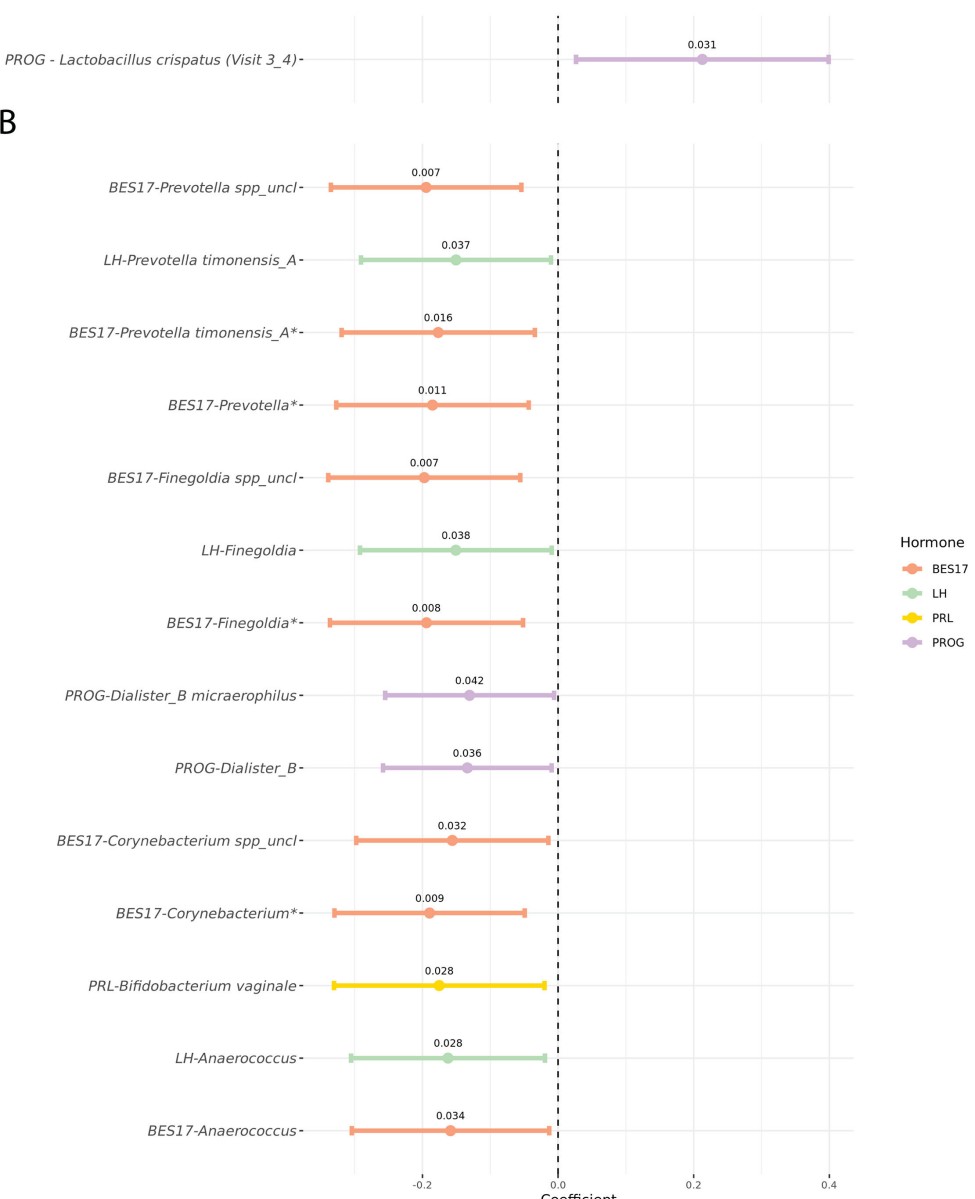

**FIG 6** Relationship between sex hormones and bacterial taxa abundances. The forest plots display the significant associations between sex hormones and bacterial taxa abundances when considering two consecutive phases (A) or in the overall menstrual cycle (B). The x-axis represents the model coefficients, indicating the strength and direction of each association, while the y-axis lists bacterial taxa names. Each line illustrates the 95% confidence interval for the coefficient, with colors representing the sex hormone associated with each taxon. The P-value for each model is shown above the corresponding point. Associations that remain significant after FDR correction, with a threshold of 0.2, are highlighted with an asterisk on the y-axis. The legend indicates the color coding for the sex hormones (BES17, 17-beta estradiol; FSH, follicle-stimulating hormone; LH, luteinizing hormone; and PRL, prolactin). Of note, some of the ASVs classified as *Bifidobacterium* spp. by the GTDB used in this study would be classified as *Gardnerella vaginalis* according to other nomenclatures (see Materials and Methods and Fig. S1).

All our volunteers had a natural menstrual cycle and did not use hormonal contraceptives for at least the past 30 days. This, together with the overall limited use of hormonal contraceptives in the Italian population (average 19.1% of females during reproductive age vs ~30% in Northern and Western European countries, according to WHO [40]) could

explain the observed low abundance of *L. crispatus*. Prolonged and/or early use of hormonal contraceptives may also be relevant. In Italy, free access to hormonal contraceptives was given only in 2023, while in some Northern European countries, it has been available for decades (41), facilitating its use in the young generation and for longer periods. Finally, differences in the abundance of the two *Lactobacillus* species may be attributed to variations in the host-immune system, in the amount and quality of vaginal secretion, in genetic variants of genes encoding receptors expressed on the surface of the vaginal epithelial cells, and other genetically determined factors specific to each host and bacteria (7).

The very high prevalence and abundance of *L. iners* in our cohort of healthy women is striking and challenges the idea that *L. iners*-dominated microbiotas are "unhealthy" or "dysbiotic." In fact, W4H volunteers have reported no current urogenital symptoms or diagnosed sexually transmitted diseases or vaginal infections, so we can rule out the abundance of *L. iners* in our samples due to either vaginal pathologies or infections. We speculate that the presence of different strains could be one of the most plausible explanations for geographic and clinical differences. Unfortunately, with the 16S sequencing approach, we (and others) are unable to identify which strains of *L. iners* are present in the studied vaginal microbiota. However, given its high genetic variability, it is reasonable to assume that certain strains of *L. iners* may promote healthy conditions, while others may increase the risk of infection, thus explaining its presence as species abundance in both healthy and dysbiotic vaginal conditions (11, 42). In addition, a woman can be colonized by multiple coexisting strains with different genetic characteristics and metabolic functions, enabling the survival of this species in a variety of vaginal microenvironments (43). A deep genetic study on *L. iners* strains coupled with human phenotypes is needed to better understand the ecology of this bacterial species and its differential impact on infections and other gynecological diseases.

Our results on dynamic changes along menstrual cycle phases agree with previous studies reporting that vaginal microbiota is generally stable over the menstrual cycle in healthy women and support current knowledge on the strong stability of microbiomes dominated by *L. crispatus* (2, 4, 19). Furthermore, we were able to pinpoint species that show dynamic changes in a minority of women, namely *L. iners*, *L. gasseri, L. jensenii,* and *Streptococcus* spp. The decreased stability conferred by these species could be explained by the production of two distinct isoforms of lactic acid by these vaginal microorganisms. Unlike *L. crispatus, L. iners*, *Bifidobacterium* spp., and *Streptococcus* spp. produce only the L-enantiomer of lactic acid, which allows easier colonization by other microbes than species also producing the D-enantiomer (5, 44). Furthermore, *L. iners* possesses additional genes compared to *L. crispatus*, encoding stress-tolerance proteins, iron–sulfur proteins, exogenous L-cysteine transport, and exhibits superior metabolic adaptation to the changing carbohydrate sources in the vaginal environment (1, 45).

Intriguingly, we observed that even though only a few women's microbiota changed along the menstrual cycle, most changes occurred from the follicular to the ovulatory phase, with most of these aligning to an *L. crispatus* profile. Beta diversity was also lower during the ovulatory and early luteal phases compared to the follicular phase, which could be due to a protective mechanism to support a potentially fertilized ovum. It is possible that the fluctuation of sex hormones plays a key role in vaginal microbiota changes. Our analyses showed that 17-beta estradiol has a major impact on longitudinal variation in taxa abundances, although none of the hormone-taxa associations involved *Lactobacillus* species, and thus they cannot fully explain the changes we observed during the ovulatory phase. Nonetheless, these associations with 17-beta estradiol are fascinating. In fact, we detected significant negative relationships with *Prevotella* spp. and *Finegoldia* and *Corynebacterium* genera, taxa that are often found in high abundances in women after menopause with menopause-associated symptoms, such as vulvo-vaginal atrophy and genito-urinary symptoms (46). The decline in estrogen levels during menopause may play a prominent role in driving microbiota change, as reduced estrogen leads to a decrease in glycogen by vaginal cells. This causes an elevated vaginal

pH, creating conditions that favor the colonization of multiple microbial species adapted to a less acidic environment, such as *Bifidobacterium* spp. and *Prevotella* spp., as well as fungi of the genus *Candida*.

Our study has several strengths and novelties, including the high-depth 16S rRNA sequencing, the enrollment of naturally menstruating women, the direct measurement of sex hormones, and the investigation of a Southern European healthy population. While our results do not have a direct clinical implication, knowledge of vaginal microbiota dynamics under physiological sex hormone variations in all populations is an essential route for investigating the potential role of microbes in the diagnosis and prevention of disease conditions such as infertility and endometriosis.

We also recognize the limitations of the study. First, we did not collect samples during menses, a stage where important changes in bacterial composition have been reported to occur, leading in some women to an excess of anaerobic bacteria (2). The compositional shifts observed for eight women between the follicular phase and ovulatory phase, five of which occurring from anaerobic to lactobacilli-dominated CSTs, may be a signature of a perturbation that occurred during menses. Still, there was no evidence in our data that these compositional shifts were due to women collecting vaginal swabs on those days right after menses (days 5 and 6) compared to women who did not experience shifts (Fisher's test $P = 0.71$). Moreover, we have focused on a single menstrual cycle, and therefore, we were unable to assess if observed changes were cyclic, i.e., the microbiota of that minority of women experiencing a shift returns to the original profile after menstruation. Furthermore, we have no measurement of vaginal pH, which is a crucial factor in the regulation of the microbiota. Finally, while we have employed a very deep 16S sequencing approach, we are limited to the relative abundance of bacteria and do not have information on bacterial function or strains, features that metagenomic sequencing could provide.

In conclusion, our study highlights the importance of extending investigations on vaginal microbiota in healthy women from diverse populations, also from the same ethnicity, and the need to assess stability and dynamics with sex hormone changes, including those occurring during a natural menstrual cycle.

## ACKNOWLEDGMENTS

We thank all the volunteers who participated in the study for their time and commitment. We thank the physicians of the University of Trieste for their contribution to the recruitment phase, including but not limited to Francesco Cracco, Roberta Maria Gentile, Elena Stefani, and Michele Stracquadaini, the directors of I.R.C.C.S. Burlo-Garofolo and IRGB-CNR for logistic support, the Agenzia Regionale Sardegna Ricerche for providing access to laboratories and technologists Davide Murrau, Marco Masala, and Michele Marongiu from the IRGB-CNR for IT support. Finally, we thank Dr. Alessandra Meloni and Dr. Ferdinando Coghe from the Azienda Ospedaliero Universitaria di Cagliari and Prof. Stefano Guerriero from the University of Cagliari for having joined the Women4Health project and initiating a second recruiting center in Cagliari starting in October 2024. Likewise, we thank Prof. Maria Raffaella Barbaro and Prof. Giovanni Barbara from the IRCCS Azienda Ospedaliero Universitaria di Bologna, Policlinico Sant' Orsola, for having joined the Women4Health project and initiating a third recruiting center in Bologna starting in April 2025.

This study was co-funded by the Italian Ministry of Health through the contribution given to the Institute for Maternal and Child Health IRCCS Burlo Garofolo, Trieste, Italy (SD 02/21 to G.G.), by the European Union (ERC Stg 2022 to S.S., acronym SEMICYCLE, GA n.101075624), by the Next Generation EU, in the context of the National Recovery and Resilience Plan, Investment PE8–Project Age-It: "Ageing Well in an Ageing Society, " (DM 1557 11.10.2022, to S.S.) and by NutrAGE grant (CNR Project FOE-2021 DBA.AD005.225, to S.S.). In addition, S.S. received a PRIN2022 grant from the Next Generation EU funds (DSB.PN004.021 2022PMZKEC_LS2_PRIN2022 SANNA, CUP B53D23008300006).

The views and opinions expressed are, however, those of the author(s) only and do not necessarily reflect those of the European Union or the European Research Council. Neither the European Union nor the granting authority can be held responsible for them.

F.D.S., G.G., and S.S. conceptualized the study. E.V., F.C., A.M., F.B., M.L.F., and S.S. wrote the original draft. S.S. supervised the project. G.G. and S.S. acquired funding. E.V., F.C., and S.S. performed statistical and bioinformatic analyses. D.V.Z., V.L.F., R.G., J.S., N.K., A.Z., and S.S. provided critical support for bioinformatic analyses. S.L., S.C., G.V.B., A.K., F.D.S., D.M., and G.G. collected samples and data. A.M., F.Cr., S.I., F.B., and M.L.F. processed vaginal samples. All authors read and revised the manuscript and approved its final version. Furthermore, all authors agree to be accountable for all aspects of the work in ensuring that questions related to the accuracy or integrity of any part of the work are appropriately investigated and resolved.

The authors declare that they have used generative artificial intelligence, specifically Quillbot, ChatGPT, and Reverso online platforms only to check spelling and grammar. The authors declare that ChatGPT was also used to improve R scripts for handling data formatting and for the generation of figures.

## AUTHOR AFFILIATIONS

[1]Institute of Genetic and Biomedical Research (IRGB), National Research Council (CNR), Monserrato, Italy
[2]Institute for Maternal and Child Health—IRCCS "Burlo Garofolo", Trieste, Italy
[3]Department of Medicine, Surgery and Health Sciences, University of Trieste, Trieste, Italy
[4]Department of Genetics, University Medical Center Groningen, Groningen, the Netherlands
[5]Department of Life Science, University of Trieste, Trieste, Italy
[6]Department of Gastroenterology and Hepatology, University Medical Center Groningen, Groningen, the Netherlands

## PRESENT ADDRESS

F. De Seta, Department of Obstetrics and Gynecology, IRCCS San Raffaele Scientific Institute, University Vita and Salute, Milan, Italy

## AUTHOR ORCIDs

E. Vinerbi http://orcid.org/0009-0001-2167-6507
F. Chillotti http://orcid.org/0009-0005-4788-5912
A. Maschio http://orcid.org/0000-0002-4238-9144
S. Lenarduzzi http://orcid.org/0000-0001-8450-1694
S. Camarda http://orcid.org/0009-0002-6946-9280
F. Crobu http://orcid.org/0000-0001-7720-8158
D. V. Zhernakova http://orcid.org/0000-0001-6531-3890
V. Lo Faro http://orcid.org/0000-0003-4931-7327
G. Beltrame Vriz http://orcid.org/0000-0002-8656-2711
S. Incollu http://orcid.org/0009-0008-0651-8421
J. Spreckels http://orcid.org/0000-0002-8711-1736
N. Kuzub http://orcid.org/0000-0001-8262-8747
R. Gacesa http://orcid.org/0000-0003-2119-0539
A. Zhernakova http://orcid.org/0000-0002-4574-0841
F. De Seta http://orcid.org/0000-0003-1611-0813
D. Mazzà http://orcid.org/0000-0003-4754-9394
F. Busonero http://orcid.org/0000-0003-0921-2045
M. L. Ferrando http://orcid.org/0000-0002-1398-8244
G. Girotto http://orcid.org/0000-0003-4507-6589

S. Sanna ⬤ http://orcid.org/0000-0002-3768-1749

## FUNDING

| Funder | Grant(s) | Author(s) |
| --- | --- | --- |
| Ministero della Salute | SD 02/21 | G. Girotto |
| European Research Council | 101075624 | Serena Sanna |
| Next Generation EU | Age-It [DM 1557 11.10.2022 ] | Serena Sanna |
| Consiglio Nazionale delle Ricerche | NutrAGE grant (CNR Project FOE-2021 DBA.AD005.225) | Serena Sanna |
| Next Generation EU | 2022PMZKEC_LS2_PRIN2022 SANNA | Serena Sanna |

## AUTHOR CONTRIBUTIONS

E. Vinerbi, Data curation, Formal analysis, Methodology, Visualization, Writing – original draft, Writing – review and editing, Software | F. Chillotti, Data curation, Formal analysis, Methodology, Visualization, Writing – original draft, Writing – review and editing, Software | A. Maschio, Data curation, Methodology, Writing – original draft, Writing – review and editing, processed biological samples | S. Lenarduzzi, Data curation, Project administration, Writing – review and editing, Investigation | S. Camarda, Data curation, Writing – review and editing | F. Crobu, Data curation, Writing – review and editing | D. V. Zhernakova, Data curation, Writing – review and editing | V. Lo Faro, Data curation, Writing – review and editing | G. Beltrame Vriz, Writing – review and editing, Data curation | S. Incollu, Data curation, Writing – review and editing | J. Spreckels, Data curation, Writing – review and editing | N. Kuzub, Data curation | A. Kadric, Data curation | R. Gacesa, Data curation, Writing – review and editing | A. Zhernakova, Data curation, Writing – review and editing | F. De Seta, Conceptualization, Writing – review and editing | D. Mazzà, Data curation, Writing – review and editing | F. Busonero, Data curation, Methodology, Writing – original draft, Writing – review and editing | M. L. Ferrando, Methodology, Writing – original draft, Writing – review and editing, Data curation | G. Girotto, Conceptualization, Funding acquisition, Supervision, Writing – review and editing | S. Sanna, Conceptualization, Data curation, Methodology, Funding acquisition, Supervision, Writing – original draft, Writing – review and editing, Investigation, Project administration, Visualization

## DATA AVAILABILITY

All our code used to analyze the data and instructions are available at https://github.com/Sanna-s-LAB/Women4Health. The human-depleted 16S rRNA and ITS sequences are deposited at the Sequence Read Archive (SRA) repository under BioProject accession number PRJNA1222832. Participant-level personally identifiable data cannot be shared with respect to participants' informed consent and are protected under the Italian Personal Data Protection Code and European Regulation 2016/679 of the European Parliament and of the Council (GDPR) that prohibit distribution even in pseudo-anonymized form. All data necessary to support the conclusions drawn in this study are available in the supplemental tables.

## ETHICS APPROVAL

The study protocol—including informed consent forms—was approved by the Friuli-Venezia Giulia Ethical Committee in September 2021 with prot. N. 0034184/P/GEN/ARCS and modifications amended in September 2023 with prot. N. 0033477/P/GEN/ARCS.

## ADDITIONAL FILES

The following material is available online.

## Supplemental Material

**Supplemental figures (mSystems00983-25-S0001.docx).** Figures S1 to S17.
**Supplemental tables (mSystems00983-25-S0002.xlsx).** Tables S1 to S18.

## Open Peer Review

**PEER REVIEW HISTORY (review-history.pdf).** An accounting of the reviewer comments and feedback.

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
