## [Reviewer comments · mSystems]

***Lactobacillus iners* dominates the vaginal microbiota of healthy Italian women of reproductive age.**

Elena Vinerbi, Fabio Chillotti, Andrea Maschio, Stefania Lenarduzzi, Silvia Camarda, Francesca Crobu, Daria Zhernakova, Valeria Lo Faro, Giulia Beltrame Vrizz, Simona Incollu, Johanne Spreckels, Nataliia Kuzub, Alma Kadric, Ranko Gacesa, Alexandra Zhernakova, Francesco De Seta, Daniela Mazzà, Fabio Busonero, Maria Ferrando, Giorgia Girotto, and Serena Sanna

Corresponding Author(s): Serena Sanna, Institute of Genetic and Biomedical Research (IRGB)

Review Timeline:

Submission Date:	July 2, 2025
Editorial Decision:	July 28, 2025
Revision Received:	July 29, 2025
Accepted:	August 1, 2025

Editor: Sarah Hird

Reviewer(s): The reviewers have opted to remain anonymous.

Transaction Report:

DOI: <https://doi.org/10.1128/msystems.00983-25>

Hello Dr. Sanna -

I'm ready to accept your manuscript and appreciate the excellent revisions but there are a few minor typographical errors that Reviewer 2 noted so I'm hoping you can correct those and upload a new final copy as soon as you're able. Don't worry about making a new response to reviewers or even a track changes document - just please upload a new "clean" final version and we'll be all set. Let me know if this is unclear - thank you!

Sarah

Re: mSystems00983-25 (*Lactobacillus iners* dominates the vaginal microbiota of healthy Italian women of reproductive age.)

Dear Dr. Serena Sanna:

Revision Guidelines

Sincerely,
Sarah Hird
Editor
mSystems

Reviewer #1 (Comments for the Author):

I truly appreciate the effort the authors have made to thoroughly address my comments and revise the manuscript accordingly. I have no further comments and am confident that this paper will make a valuable contribution to the field.

Reviewer #2 (Comments for the Author):

The authors have thoughtfully responded to the comments, corrected errors, and provided clarification where appropriate. Major concerns have been addressed.

Further clarification on the classification of *Gardnerella vaginalis* as *Bifidobacterium* by the GTDB could be added by parenthetically noting that *Bifidobacterium* spp. or *Bifidobacterium vaginae* could be classified as *Gardnerella* the first time they are mentioned in the results, and possibly in figure legends where appropriate.

Additional minor copyediting issues should be addressed throughout, including the following:

- Line 61: "Microbiota" is plural. The phrase, "a 'healthy' microbiota" should be either "'healthy microbiota" or "a 'healthy' microbiome"
- Line 85: Fannyhessea is misspelled
- Line 541: pinpointto should be pinpoint to

The authors have thoughtfully responded to the comments, corrected errors, and provided clarification where appropriate. Major concerns have been addressed.

Further clarification of the classification of *Gardnerella vaginalis* as *Bifidobacterium* by the GTDB could be added by parenthetically noting that *Bifidobacterium spp.* or *Bifidobacterium vaginale* could be classified as *Gardnerella* the first time they are mentioned in the results.

Additional minor copyediting issues should be addressed throughout, including the following:

- Line 61: “microbiota” is plural, the phrase, “a ‘healthy’ microbiota” should be either “‘healthy microbiota” or “a ‘healthy’ microbiome”
- Line 85: *Fannyhessea* misspelled
- Line 541: pinpointto should be pinpoint to

RESPONSE LETTER

We would like to thank all reviewers for their constructive suggestions. Below is a point-by-point response to the concerns raised and how these have been addressed.

Reviewer #1 (Comments for the Author):

This manuscript describes a study investigating the vaginal microbiota composition across different phases of the menstrual cycle in 61 healthy Italian women. The primary finding is the dominance of *Lactobacillus iners*, which contrasts with some previous European studies that have suggested a pathogenic role for this species. The study suggests potential differences in vaginal microbiota profiles across European populations.

Abstract

- Line 43: write *Lactobacillus iners* in full (first time)
- Line 48-49: be more specific on what can of impact

We have addressed these points in the abstract. At lines 48-49 we rephrased the sentence with the following:

“Finally, using linear mixed models we assessed the association between taxa relative abundance and five sex-hormones along the menstrual cycle. Among these, 17-beta estradiol showed the largest number of significant associations, linking its increase to a decrease in relative abundance of taxa that are more common after menopause.”

Importance

- Line 64: what is 'normal'? Can specify that you mean normal in this kind of population of x age & generally healthy?

We have followed the reviewer suggestion rephrasing the sentence as follows:

“Our findings help define what constitutes a “healthy” microbiota in generally healthy Italian women of reproductive age and may inform future strategies for diagnosing or preventing women’s health conditions.”

Introduction

- Line 74: now 4 references from the same research group that in general say the same. Try to diversify, perhaps with European studies

This comment concerns the sentence “Producing lactic acid and antimicrobial molecules, these bacteria reduce the risk of colonization by other microorganisms, including potential pathogens involved in infections and diseases of the vaginal tract (France, Mendes-Soares, and Forney 2016); (Ravel et al. 2011); (Gajer et al. 2012); (Holm, France, et al. 2023)”

We have followed the reviewer’s suggestion and added the following references:

1. Petrova et al 2025, “*Lactobacillus species as biomarkers and agents that can promote various aspects of vaginal health*” (<https://doi.org/10.3389/fphys.2015.00081>); this is a review article that discusses the beneficial effect of *Lactobacillus* species
2. Sai Ravi Chandra Nori et al 2025 “*Strain-level variation among vaginal Lactobacillus crispatus and Lactobacillus iners as identified by comparative metagenomics*” (<https://doi.org/10.1038/s41522-025-00682-1>); in this research article, *L.iners* and *L.crispatus* strain-specific genetic components associated with metabolic output, colonization and host-microbe interactions are described
3. Scillato et al 2021, “*The aggregation of three Lactobacillus strains (L. gasseri, L. fermentum, and L. crispatus) inhibits the colonization of other pathogens, particularly multidrug-resistant uropathogens*” (<https://doi.org/10.1002/mbo3.1173>); in this research article, in-vitro assays are used to show antimicrobial activity of *Lactobacillus* strains against urogenital pathogens

- Line 75-76: not necessarily - this needs a major consideration because it does not stay stable during pregnancy & infections that will happen to most women at least once. Best to make that sidenote clear here

We thank the reviewer for this suggestion. Since during pregnancy fluctuations are observed but on species levels, we replaced the word *species* with *genus* and clarified the expected variations during infection. The sentence now reads as follows:

“In general, the prevalence of the Lactobacillus genus in healthy women increases with puberty and remains stable until menopause, except during vaginal infections”

- Line 80: 'from' instead of 'by'

We fixed this error.

- Line 83: add reference

The correspondence with *G. vaginalis* and *Bifidobacterium spp.* is yet a matter of debate. Named *G. vaginalis* according to NCBI nomenclature, this species is instead classified as *Bifidobacterium vaginae* and also other *Bifidobacterium spp.* according the Genome Taxonomy Data Base (GTDB) nomenclature

<https://gtdb.ecogenomic.org/searches?s=al&q=gardnerella%20vaginalis>

We have identified the following reference as best research article to describe the multiple matches of *Gardnerella vaginalis* with *Bifidobacterium vaginae* as well as other species and included it on the manuscript text.

Cornejo O. et al 2017 “Focusing the diversity of Gardnerella vaginalis through the lens of ecotypes” (<https://doi.org/10.1111/eva.12555>).

In our study, we used the GTDB nomenclature and thus *G. vaginalis* does not appear among the identified species. Since to define CTSs we run the VALENCIA algorithm, which uses *G.vaginalis* ASVs, we repeated taxonomy classification using the SILVA database, which follows the NCBI nomenclature. We then used in the VALENCIA algorithm only on those ASVs that exactly matched with *G. vaginalis* in SILVA. This step is now better described on the manuscript methods session, and we also added a Supplementary Figure in support (Figure S1).

The text now reads

“In addition, we classified our samples according to the Community State Types (CSTs) definition of the VALENCIA algorithm (France et al. 2020) (see URLs). Toward this end, it was necessary to manually modify the taxonomy names according to those required by VALENCIA (Table S6). To identify which ASV could be attributed to *Gardnerella vaginalis*, given the different nomenclature in GTDB, we run again DADA2 with the SILVA v.138.1 database where the nomenclature of species aligns to that used in the VALENCIA algorithm (**Figure S1**). Specifically, we identified 46 ASVs assigned to the genus Bifidobacterium using GTDB v r.95, among these, 15 ASVs classified as B. vaginalis (6 ASV) and B. unclassified (9 ASV) corresponded to Gardnerella vaginalis when annotated using the SILVA database. These 15 ASVs were renamed accordingly, while the remaining ASVs were discarded, to use a VALENCIA algorithm.”

- Line 84: example of some pathological conditions

We have added two examples and updated the references in the sentence accordingly.

- Line 93: "it is"

We fixed this error.

- Line 94: add reference Lebeer 2023 on co-existence of *L. iners* and *L. crispatus*

We added the reference.

- Line 107-108: rephrase for clearer reading + check if this reference is indeed relevant for this sentence (can also go without)

We removed the reference and rephrased the sentence which now reads as follows:

"It is worth noting that none of these studies have characterized the association between compositional changes in the microbiota and specific fluctuations in sex hormones."

- Line 112: gut microbiome mentioned but not discussed in paper? Option for gut-vagina axis discussion in paper?

Women4Health is a large project that includes the collection of several biological samples. Recruitment is still ongoing, and at this stage we have analyzed data on vaginal samples and hormonal data for the 61 women firstly enrolled. Processing of gut metagenomic data is still ongoing. To avoid confusion in the reader, we removed the details on fecal samples collection. The sentence now reads:

"They were then scheduled for 4 weekly follow-up appointments to hand in self-collected biological samples, including a vaginal swab, and to undergo a blood withdrawal. Moreover, they were given an additional questionnaire to collect information related to their health status and lifestyle in days preceding the appointment."

Materials&Methods

- Line 121: please explain in text why it here says 300 but in the rest 61 women? Was there drop-out? Or bad quality?

The recruitment of the Women4Health cohort is still ongoing. Here we presented the results from the firstly 61 enrolled volunteers. We modified the text as follows:

"In this study, we analyzed vaginal swabs, metadata, and sex hormone levels for the first 61 women enrolled in the Women4Health (W4H) cohort, an ongoing short-term longitudinal study aiming to recruit up to 300 healthy women and to follow them up through a natural menstrual cycle. The W4H study design has been previously described (Busonero et al. 2024)."

We also included a new figure, Figure1, to also visualize sampling scheme of the data included in this work, as suggested by this reviewer.

- Line 155 + Line 184-185: average bacterial and fungal reads per sample after sequencing & QC? So, separate for bacterial & fungal reads

We now report separate statistics for average number of bacterial and fungal reads at the end of the Quality Control process paragraph.

- Do you have a figure displaying the read of negative controls compared to the rest of the samples?

We now included information on the reads detected for positive and negative controls. This information has been included as Supplementary Table S2.

- Line 221: Gardnerella vaginalis

We fixed this error.

- Line 320: shotgun metagenomic sequencing?

We have renamed the paragraph "Shotgun metagenomic sequencing".

Results

- Line 340-341: can you put a percentage?

We followed reviewer suggestion and used percentages. The sentence now reads:

L. iners was detected in 43.4% of the samples (corresponding to 62.2% of women), whereas *L. crispatus* was detected in 34.9% (55.7% of women).

- What is the difference between H and D in figure S3 and explain for reader?

We modified the legend with an explanation of the two indices (now figure S4)

- Line 344: peculiar? Most papers report after the menstrual bleeding a flushout with more anaerobic bacteria, have you not seen this in your data? Good to include in discussion

We thank the reviewer for this comment. Our observations excluded the menstruation period (see newly included Figure 1 which graphically describes the sampling scheme). Vaginal swab collection starts from the follicular phase; therefore, by not sampling during

menses, we may have missed the days with the large perturbations due to pH changes that have been described in other studies.

We took a closer look to the women who experienced compositional shifts from the follicular to ovulatory phase. The first follow-up visit at the follicular phase is scheduled between day 6 to day 9 since the last menstruation (with swabs that could be collected the same day of follow-up visit or in the previous day). A total of 5 out of the 8 are shifts from CST-IV (characterized by anaerobic bacteria) toward CST-I and CST-II, the remaining shifts are from CST-III to CST-I. We noted that these 5 women (62.5%) who experienced compositional shifts from the follicular to ovulatory phase have collected vaginal swabs between day 5 and 6 (instead of between day 7 and 9); however this is not significantly different from the percentage of women who did not experienced shifts between these two phases (52% out of 52 women, Fisher test pvalue= 0.71). Therefore, while some of these shifts may reflect perturbations occurred during menses, we have not evidence for an overall signature of menstruation.

We have now rephrased the sentence as follows:

“Overall, vaginal microbiota remained stable throughout the four menstrual cycle phases here investigated”

and in the Discussion section, we further expanded the paragraph where we discuss about this point, with the following text:

“We also recognize the limitations of the study. First, we did not collect samples during menses, a stage where important changes in bacterial composition have been reported to occur, leading in some women to an excess of anaerobic bacteria [Hugarth 2024]. The compositional shifts observed for 8 women between the follicular phase and ovulatory phase, five of which occurring from anaerobic to lactobacilli dominated CSTs, may be a signature of a perturbation occurred during menses. Still, there was no evidence in our data that these compositional shifts were due to women collecting vaginal swabs on those days right after menses (day 5 and 6) compared to women who did not experienced shifts (Fisher test $p=0.71$)”.

We also updated Figure 3A (now figure 4A) showing CST shifts across menstrual cycle phases in the t-SNE plot. We removed labels and changed arrows colors to improve readability.

- Line 350: not significant?

We followed reviewer suggestion and replaced “These changes were small” with “These changes were not significant”

- Line 354: which aspects/covariates did you include in surveys?

In the surveys, we included several questions regarding sexual habits, use of medication, past pregnancies, past use of hormonal contraceptives, demographic and anthropometric parameters and smoking, all these aspects were reported and described in table S7. These were all evaluated in the context of beta diversity in the paragraph "Impact of host factors". Unfortunately, we do not have information regarding hygiene products, nor regarding pH, as indicated in the Discussion.

- Line 359-360: did the women in whom you found *Candida* have VVC or show symptoms of a fungal infection?

We thank the reviewer for this question. All women who reported known vaginal infections or urogenital symptoms were excluded at enrollment. We have now added more details on the amount of reads we detected for *Candida albicans*. We believe our observations are in line with the common knowledge that this fungus could be present at low abundance in physiological conditions and thus not a sign of current (asymptomatic) infection. Specifically, we have added the following sentence:

"The average number of reads on the 8 women with presence of Candida albicans was low (749 compared to 2832 on Lactobacilli spp. on the same women) and thus the presence of this fungus is unlikely to be a sign of infection. Of note, we excluded at enrollment all women who reported to currently have vaginal infections or urogenital symptoms"

- Line 379-380: not clear, did you compare vaginal collection before & after feces sampling from the same people?

We apologize for the confusion generated by the sentence, which was probably too concise. Women were asked to fill out a questionnaire indicating the time of day they collected vaginal swabs, and if they were taken prior to or following the collection of fecal samples. We therefore evaluate the fraction of beta diversity variability that these variables explain in a PERMANOVA analysis. To understand which major *Lactobacillus* spp. these variables were correlated to, we used a visualization approach. We compared the PCoA plots with points colored on the left side according to the most common *Lactobacillus* spp. in the sample, and in the right side according to the values of the variables derived from the questionnaire. We have clarified this in the legends of Figure S10-S11-S12, and rephrased the sentence which now reads:

"Intriguingly, using a PERMANOVA test and projecting values in the PCoA space, we observed that the microbiota of women who had at least one pregnancy before participating

in the study had a lower likelihood to be dominated by *L. crispatus* ($p=0.005$), in line with observations from the ISALA study, suggesting the existence of a potential lifelong pregnancy signature (Figure S9). Likewise, using PERMANOVA we observed that the microbiota of older women is less likely dominated by *Lactobacillus* spp. ($p=0.003$), in line with the observed decline in their abundance during menopause (Figure S10) (Lebeer et al., 2023)(Muhleisen & Herbst-Kralovetz, 2016) Similarly, dominance of *Lactobacillus* spp. is less frequent in women who reported to have collected vaginal swabs after feces collection compared to those who collected them prior feces ($p=0.02$) (Figure S11), and in those who collected vaginal swabs in the morning compared to those who collected them during the day ($p=0.003$) (Figure S12).”

- Line 384: define Northern European women (which countries & add sample size + reference).

We added information on samples size and country in Table 1. We have also added countries and references to the text. We defined Northern European those studies carried out in Belgium, Denmark and Sweden. In Table 1 we also reported a study carried out in France, but we did not define this study as Northern European as they included 70% of women from Southern France. We elaborated the findings from the French study in the Discussion.

- Line 396: *L. iners* in italic
- Line 400: microbiota in itself in already plural
- Line 415: *jensenii* between capital

We fixed all these errors.

- General comment: I believe you should also add a dot after spp such as *Lactobacillus* spp
We fixed this in all occurrences.

- Line 439: possible to identify the unclassified with shotgun?

We tried to identify the unclassified *Streptococcus* species using 10 samples for which WGS data was available. We found that only three *Streptococcus* species (*S. agalactiae*, *S. pyogenes*, and *S. anginosus*) were present and they appeared in just two samples (P1S61 and P3S206). Next, we looked at these two samples in the 16S data and selected the same three *Streptococcus* species, along with the *Streptococcus* spp. unclassified. Based on the available samples, we concluded that it is not possible to determine which species the unclassified *Streptococcus* corresponds to using WGS data.

Below is a plot showing their relative abundances.

- Fig 16S: is there a way for a more compact figure? Perhaps heatmaps? (also good to have a legend for letter abbreviations)

Figure S16 (now figure S17) is an expanded version of Figure 5B (now figure 6B), where we show the coefficient of the hormone-bacteria association model across all time points. In Figure S16 (now figure S17), we show scatterplots and correlation lines for each bacteria-hormone pair within each time point as a sanity check. An heatmap will only display the correlation value and thus would not be sufficient to fully show relationships.

If this Figure S17 is too long for Editorial purposes, we will be happy to only include an heatmap and provide expanded figures on our GitHub page. Please note that this figure has been updated to add unit scale of sex hormone levels on the Y axis.

- Line 446-447: what kind of changes? More or less?

We apologize for the use of general wording. We now provide directions of changes according to coefficients displayed in Figure 5 (now Figure 6).

Specifically, the sentence now reads:

“We found that an increase in abundance of L. crispatus between early luteal and late luteal phases was associated with an increase in progesterone (PROG) levels ($p=0.03$) (Table S16 and Figure 6A).”

- Lines 450-460: please be more specific what do you mean with changes & associations: is there an increase/decrease when there is more or less of a particular hormone?

We apologize once more for the use of general wording. We now provide directions of changes according to coefficients displayed in Figure 5.

Specifically, the sentence now reads:

“All these associations were negative, indicating that an increase in sex hormones levels is associated with a decrease in the relative abundance of these bacteria (Figure 6B and Figure S17).”

Discussion

- Line 474-475: can you give a hypothesis?

We appreciate the fact that even the reviewer is intrigued by this result. In the subsequent paragraph, we attempted to formulate hypotheses to explain this difference. Among many, we mentioned the potential prolonged use of hormonal contraceptives in Northern Europe compared to Southern, variations could certainly influence the composition of the microbiota. In Italy, free access to hormonal contraceptives was given only in 2023, while in North European countries have been available since decades, facilitating its use in the young generation and for longer periods. We have added this to the Discussion session

Furthermore, among the collected variables in the Women4Health cohort, we also have daily diet records. We attempted to investigate the potential impact of diet using the 3-days average intake of several macronutrients (sugar, cholesterol, fat, fibers, carbohydrates, protein) and total calories, but none of these associated with high abundance (>30%) of *L.iners* or *L.crispatus*. Given the limited number of samples to reach sufficient statistical power to investigate specific foods, we have not included this part in the manuscript.

We report here below a figure with results for the reviewer. The figure is a forest plot with bars showing the coefficient of diet intake (and the pvalue of top) for the logistic mixed model which takes into account age and BMI

Model1: $\text{Logit}(L. iners \geq 30\% / L. iners < 30\%) \sim \text{age} + \text{BMI} + \text{nutrient_intake} + (1 | \text{ID})$

Model 2: $\text{Logit}(L. Crispatus \geq 30\% / L. Crispatus < 30\%) \sim \text{age} + \text{BMI} + \text{nutrient_intake} (1 | \text{ID})$

- Line 489: *L. crispatus** and line 499 & 524
- Line 494: space missing *L. iners*

We fixed all these errors.

- Line 499-502: reference?

We have added reference Chandra Nori et al NPJ Biofilms and microbiome 2025 in support of the statement regarding different host-microbiome interactions at strain-level in *L. crispatus* and *L. iners*

- Line 505: a new result to be mentioned in results?

The referee refers to the sentence in the Discussion “In fact, W4H volunteers have reported no current symptomatic vaginal infections, so we can rule out the abundance of *L. iners* in our samples due to either vaginal pathologies or infections.” We have added this information in the Results section to further emphasize that our volunteers are healthy and free from urogenital infections or symptoms.

Specifically, the following sentence was added:

“Notably, all volunteers have reported absence of urogenital symptoms or diagnosis of sexually transmitted diseases or infections, so we can rule out the abundance of L. iners in our samples due to either vaginal pathologies or infections.”

- Line 531: microbiota is already a plural

We fixed this error

- Line 546: such as?

We rephrased the sentence as follows:

“This causes an elevated vaginal pH, creating conditions that favor the colonization of multiple microbial species adapted to a less acidic environment, such as Bifidobacterium spp. and Prevotella spp., as well as fungi of the genus Candida.”

- A graphical overview of the sampling would be beneficial

We have now included a graphical overview of the sampling in the Women4Health cohort as Figure 1.

Reviewer #2 (Comments for the Author):

Summary

In the present study, Vinerbi, Chillotti, Maschio, Lenarduzzi et al. measure the vaginal microbiome during four phases of the menstrual cycle and associations with hormone levels. The authors describe the relative stability of the vaginal microbiome, especially when dominated by *L. crispatus*, as has been documented previously. The authors also show a negative association between 17-beta estradiol and several non-Lactobacillus species. The study does not include measurements of vaginal pH, which is an important factor in the vaginal microbiome, and does not include measurements during menses, but the authors note these potential limitations. Overall, the study adds to the literature regarding vaginal microbiome changes during the menstrual cycle, but some revisions could improve clarity and completeness of the analyses.

Major Comments

The authors suggest that high abundance of *L. iners* in this cohort challenges its proposed pathogenic role. High abundance of *L. iners* alone may not preclude it from a potentially pathogenic role. Although the study may suggest it is common in this specific cohort, comparisons to symptoms should be included to describe a potential role, or lack thereof, in pathogenesis. Such symptoms could include pH or other Amsel's criteria, Nugent's score, or other symptoms of discomfort. In the present study, the authors note that the *L. crispatus*-dominant microbiomes were the most stable.

We thank the reviewer for this comment. There is not clear evidence today for a pathogenic role of *L. iners* supported by in vitro or in vivo studies, but rather only associations showing higher frequency in women with vaginal conditions. In our study, we focused solely on women who reported the absence of any vulvovaginal symptoms. Women reporting having a current infection or urogenital symptoms or have received a diagnosis of it, were not

enrolled. Therefore, no correlation with symptoms can be performed. To further strength our selection criteria, we also added the following question to each follow up visit survey: “Did you experience any difficulty or pain during vaginal swab collection?”. Only two women have replied yes, and only at one time point. None of their microbiota samples showed a higher prevalence of *L. iners*. One had a heterogeneous composition in which the ten most abundant taxa did not include *L. iners*, and the other was mostly dominated by *L. crispatus* (relative abundance 92%).

We have made clear in the Discussion and in the Results section that all our women are healthy and have not reported any symptoms, supporting the hypothesis the *L. iners* can be highly dominant in healthy vaginal microbiota.

We did not perform any objective measurements such as pH evaluation, Amsel criteria, or Nugent score, due to the existing debate in the literature regarding asymptomatic vaginal dysbiosis (e.g., altered pH, Nugent score 4–6), and whether these should be considered as physiological states or asymptomatic infections (Muzny et al 2020; 10.1007/s11908-020-00740-z).

Based on Figure S1, *Gardnerella* is the most abundant genus in some samples and the second most abundant genus in many samples. However, *Gardnerella* appears to be included within the "Other" category in the stacked bar plots. Why is this? It appears that these plots list the 10 most abundant species, but they should instead reflect the most abundant taxa. *Gardnerella* often cannot be defined to the species level by variable regions of the 16S rRNA gene. Perhaps adjust the criteria for inclusion in the bar plots to abundant species or genera.

We apologize for making a mistake in the legend of Figure S1 (now Figure S2). We have now fixed it. For this, we would like to point out that genus *Gardnerella* is called *Bifidobacterium* in this manuscript since we have used the GTDB (instead of the less updated SILVA database) which uses this nomenclature. We have clarified this in the Introduction.

Furthermore, in the Methods section, we now better describe how we treated the single *Bifidobacterium* ASVs and renamed some of these as *Gardnerella vaginalis* for the VALENCIA algorithm. We also included a figure (Figure S2) with a schematic match of Avs between the GTDB and SILVA database. The paragraph in the Methods section now reads:

“To identify which ASV could be attributed to Gardnerella vaginalis, given the recent change in nomenclature, we used DADA2 with the database SILVA v.138.1 where the nomenclatures of species align to those used in the VALENCIA algorithm (Figure S2). Specifically, we identified 46 ASVs assigned to the genus Bifidobacterium using GTDB v r.95, among these, 15 ASVs classified as B. vaginalis (6 ASV) and B. unclassified (9 ASV) corresponded to Gardnerella vaginalis when annotated using the SILVA database. These 15 ASVs were

renamed accordingly, while the remaining ASVs were discarded, to use a VALENCIA algorithm. “

Lines 49-52: It may improve clarity here to specify that 17-beta estradiol had the greatest number of statistically significant associations with species, and that these were negative associations.

We modified the text accordingly.

Line 133: The methods describe the days within the menstrual cycle when follow-up visits were scheduled to align with specific phases. As vaginal swabs were self-collected on the day of or prior to these appointments, were any vaginal swabs collected the day before the described windows, i.e. day 17 or 23 of a cycle? If so, would this impact results?

We thank the reviewer for this comment. This out-of-range collection occurred only 11 times out of 212 time points, corresponding to 8 women out of 61 studied. Therefore, the impact is likely minor for the longitudinal model, and if any would only decrease power (by reducing variability across time points) without increasing false positives. For the linear model with sex hormones data, we expect the impact would be even more negligible since we rely on directly measured sex hormones rather than days since menstruations started.

Line 423 and Figure S14: In panel S14a, CST-V samples represent only 2.5% of all samples. However, based on S14b, at least ~14% of samples within specific phases are CST-V. How is this possible? We checked the CST values and found an internal error in the script.

We thank the reviewer for this comment. We indeed found out that our previous plot and calculations were not correct. We have now updated the numbers in the text as well as in the figure (now figure S16)

Line 452: Why were Lactobacillus and Streptococcus specifically chosen for assessing corresponding hormone changes? If it is because they were the most abundant, why not Bifidobacterium which was also listed as the most abundant genus?

Studying association between all consecutive time points would be inefficient given the large numbers of tests to be performed and the limited sample size. We therefore decided to test only those bacteria that showed themselves to be significant in the model bacteria ~ visit. We modified the sentence to clarify this, and now reads:

“We investigated whether these observed significant changes in Lactobacillus and Streptococcus species between two consecutive phases were associated with changes in sex hormone levels”.

Lines 442-450 and 459-470: Was *Gardnerella* assessed here? As noted above, it was among the most abundant genera, and *G. vaginalis* was highlighted as an important species in CST-IV microbiomes at line 425.

In lines 442-450 we only considered species that were significant in the longitudinal temporal model. In lines 459-470 we analyzed all taxa with a prevalence >20%, thus including *Bifidobacterium spp.* and *Bifidobacterium vaginale* that corroborate *G. vaginalis*.

We have clarified this in the text.

Minor Comments

Variations of the term "17-beta estradiol" are used throughout. For example, it is referred to as "17-beta estradiol" at line 49, "beta17-estradiol" at line 137, and "beta-17 estradiol" at line 462. Should this be consistent throughout the paper?

We apologize for the inconsistencies. We have now replaced all occurrences with 17-beta estradiol.

In Figure 2, it would be helpful if the panels were labeled with the corresponding phase during sampling.

We modified the figure accordingly.

Line 81: If using the word dysbiotic, it should be clearly defined. There is also a debate regarding whether this is an accurate classification of microbiome states (e.g. Olesen & Alm 2016, Dysbiosis is not an answer; Nature Microbiology)

We modified the text and used the word “unhealthy microbiome” instead of dysbiotic.

Line 226: The word "vaginalis" should be lowercase in *Gardnerella vaginalis*, and *Gardnerella* is misspelled here.

Lines 345, 349, and others: A period is missing after the "spp" in *Lactobacillus spp.* This occurs after other genus names throughout as well.

Lines 416-417: I suggest referring to "this cohort" instead of "the Italian population" here.

Lines 509 and 535: The word "crispatus" should be lowercase in *L. crispatus*. Done in text

For all the above, we modified the text accordingly.

Re: mSystems00983-25R1 (*Lactobacillus iners* dominates the vaginal microbiota of healthy Italian women of reproductive age.)

Dear Dr. Serena Sanna:

Your manuscript has been accepted, and I am forwarding it to the ASM production staff for publication. Your paper will first be checked to make sure all elements meet the technical requirements. ASM staff will contact you if anything needs to be revised before copyediting and production can begin. Otherwise, you will be notified when your proofs are ready to be viewed.

Sincerely,
Sarah Hird
Editor
mSystems